# Inhibition of SFTSV replication in humanized mice by a subcutaneously administered anti-PD1 nanobody

Mengmeng Ji [1,7], Jiaqian Hu [2,7], Doudou Zhang [2], Bilian Huang[2], Shijie Xu[2,3], Na Jiang[2], Yuxin Chen [4✉], Yujiong Wang [1✉], Xilin Wu [2,5✉] & Zhiwei Wu [1,2,5,6✉]

## Abstract

**Severe fever with thrombocytopenia syndrome (SFTS) is a life-threatening disease caused by a novel bunyavirus (SFTSV), mainly transmitted by ticks. With no effective therapies or vaccines available, understanding the disease's mechanisms is crucial. Recent studies found increased expression of programmed cell death-1 (PD-1) on dysfunctional T cells in SFTS patients. However, the role of the PD-1/programmed cell death-ligand 1 (PD-L1) pathway in SFTS progression remains unclear. We investigated PD-1 blockade as a potential therapeutic strategy against SFTSV replication. Our study analyzed clinical samples and performed in vitro experiments, revealing elevated PD-1/PD-L1 expression in various immune cells following SFTSV infection. An anti-PD-1 nanobody, NbP45, effectively inhibited SFTSV infection in peripheral blood mononuclear cells (PBMCs), potentially achieved through the mitigation of apoptosis and the augmentation of T lymphocyte proliferation. Intriguingly, subcutaneous administration of NbP45 showed superior efficacy compared to a licensed anti-PD-1 antibody in an SFTSV-infected humanized mouse model. These findings highlight the involvement of the PD-1/PD-L1 pathway during acute SFTSV infection and suggest its potential as a host target for immunotherapy interventions against SFTSV infection.**

**Keywords** SFTSV; Nanobody; NbP45; PD-1 Blockade; Subcutaneous Injection
**Subject Categories** Immunology; Microbiology, Virology & Host Pathogen Interaction

## Introduction

Severe fever with thrombocytopenia syndrome (SFTS) is a life-threatening infection caused by a novel bunyavirus identified as SFTS virus (SFTSV). This enveloped RNA virus, later reclassified as Dabie bandavirus, contains tripartite, single-stranded, negative-sense, and ambisense genomes, comprising three distinct RNA segments—Large (L), Medium (M), and Small (S) (Xu et al, 2021a). Between 2007 and 2010, patients with unexplained infectious hemorrhagic fever in Henan and Hubei provinces of China were reported and the pathogen identified from serum samples was known as SFTSV (Stone, 2010). In the last 10 years, the endemic area of SFTS is expanding. Ticks are implicated as the prominent vectors for transmission while the human-to-human transmission was reported through contact with blood or body fluids of SFTS patients (Chen et al, 2019; Lin et al, 2020; Niu et al, 2013). The major clinical manifestations of SFTS include high fever, thrombocytopenia, leukopenia, bleeding and gastrointestinal symptoms and multi-organ dysfunctions, with case mortality rate of 6–30% (Liu et al, 2014; Tran et al, 2019; Yu et al, 2011). To date, no effective therapies, or vaccines for SFTS are available, and the mechanisms of disease pathogenesis remain unclear. SFTS has been listed as one of the important infectious diseases that endanger public health and require priority in research and intervention by the World Health Organization (WHO) in 2018.

Previous studies revealed that virus-induced cytokine storm and impairment of innate immune response were one of the important pathogenic manifestations in SFTSV infection (Song et al, 2017; Zhang et al, 2019). In addition, SFTSV could evade antiviral immunity by modulating adaptive immune response (T- and B cells) (Song et al, 2018; Wang et al, 2022). We showed that the disruption of viral Gn-specific IgG secreting B-cell differentiation and maturation was a major factor in SFTS pathology and death. The inhibition of $T_{fh}$ activation, depletion of monocytes and disruption of mDCs maturation during infection were responsible for the disruption of viral-specific B-cell differentiation and maturation (Song et al, 2018). PD-1/ PD-L1 is an important inhibitory checkpoint signaling pathway that inhibits immune responses against cancers and viral infections (Davar et al, 2021; Uldrick et al, 2022). In fatal Ebola virus infection, CTLA-4 and PD-1 were highly expressed in CD4$^+$ and CD8$^+$ T cells with dysregulation of the T-cell response (Ruibal et al, 2016). Li et al reported that PD-1 expression on dysfunctional CD4$^+$ and CD8$^+$ T cells increased significantly during the acute phase of SFTSV infection, while the role of PD-1 in the disease pathogenesis was not extensively explored (Li et al, 2018). These studies suggest that the

[1]School of Life Sciences, Ningxia University, Yinchuan, China. [2]Center for Public Health Research, Medical School, Nanjing University, Nanjing, China. [3]Y-Clone Medical Science Co. Ltd., Suzhou, China. [4]Department of Laboratory Medicine, Nanjing Drum Tower Hospital, Medical School, Nanjing University, Nanjing, China. [5]Jiangsu Key Laboratory of Molecular Medicine, Medical School, Nanjing University, Nanjing, China. [6]State Key Laboratory of Analytical Chemistry for Life Science, Nanjing University, Nanjing, China. [7]These authors contributed equally: Mengmeng Ji, Jiaqian Hu. ✉E-mail: yuxin_chen2015@163.com; wyj@nxu.edu.cn; xilinwu@nju.edu.cn; wzhw@nju.edu.cn

PD-1/PD-L1 immune checkpoints may suppress anti-SFTSV immunity and serve as potential therapeutic targets for SFTSV infection.

In this study, we examined the expression of PD-1/PD-L1 in immune cells derived from individuals diagnosed with SFTS. Moreover, we provided compelling evidence demonstrating that in vitro infection with SFTSV can induce upregulation of PD-1/PD-L1. To determine the role of PD-1/PD-L1 in the progressive disease of SFTS, we identified a new anti-PD-1 nanobodies, named NbP45 that was isolated from an alpaca immunized with human PD-1. We subsequently ascertained that NbP45 possesses the ability to suppress SFTSV replication both in vitro and in our previously reported NCG-HuPBL mice, a humanized mouse model containing components of the human immune system. Furthermore, we compared the efficacy by the different administrations of NbP45 or Tislelizumab (a licensed commercial anti-PD-1 antibody drug) in treating SFTSV infection in humanized mouse model. The pharmacokinetics and tissue distribution via different administrations were also examined. Finally, the protective mechanism of NbP45 inhibiting SFTSV replication was investigated.

# Results

## PD-1/PD-L1 was upregulated in the immune cells of SFTS patients

To determine the dynamics of PD-1/PD-L1 expression on immune cells following infection with severe fever with thrombocytopenia syndrome virus (SFTSV), peripheral blood mononuclear cells (PBMCs) were obtained from 15 SFTSV-infected patients and healthy individuals as controls (Fig. 1A,B). Flow cytometry analysis was then conducted to evaluate the expression levels of PD-1 and PD-L1 on T/B lymphocytes and monocytes. Remarkably, $CD4^+$ $CD3^+$ T cells, $CD8^+$ $CD3^+$ T cells, B cells ($CD19^+$ $CD3^-$), and monocytes ($CD14^+$ $CD3^-$) demonstrated significantly higher PD-1 expression compared to the healthy control group (Figs. 1C–F and EV1A–D). Moreover, SFTS patients displayed increased expression levels of PD-L1 on B cells and monocytes relative to the healthy control group (Figs. 1G,H and EV1E,F). To explore the relationship between the serum viral load in SFTS patients and the expression of PD-1/PD-L1 on immune cells, a comprehensive analysis was conducted and presented in Fig. EV2A–F. Notably, a positive correlation was observed between PD-1 expression in $CD4^+$ T cells (Fig. EV2A) and PD-L1 expression in $CD14^+$ monocytes (Fig. EV2F) with the viral load.

## SFTSV infection induced PD-1/PD-L1 upregulation on T/B lymphocytes

To explore the kinetics of PD-1/PD-L1 in $CD4^+$, $CD8^+$ T, and $CD19^+$ B cells during SFTSV infection, PBMCs from healthy individuals were infected with SFTSV (subtype E, JS-2013-14) (MOI = 1) and analyzed by flow cytometry for phenotypic markers at various time points. We showed that the PD-$1^+$ $CD3^+$ T cells increased gradually from 24 h post infection and reached significantly higher level after 72 h (Fig. 2A). We then specifically analyzed the expression of PD-$1^+$ of $CD4^+/CD8^+$ T cells at 72 h post infection, compared with the uninfected control, and showed

that the percentage of $CD4^+$ T cells expressing PD-1 was gradually increasing and markedly increased at 120 h (Fig. 2B), and the percentage of $CD4^+$ of $CD3^+$ T cells significantly decreased in 72–96 h post-SFTSV infection (Fig. EV3A). In addition, PD-$1^+CD8^+$ T cells significantly increased after 72 h post-SFTSV infection (Fig. 2C), while the expression of $CD8^+$ of $CD3^+$ T cells significantly decreased in 72–96 h post infection (Fig. EV3B). Interestingly, the expression of PD-$1^+$ $CD19^+$ lymphocytes increased gradually from 12 h post infection and reached significantly higher level after 72 h (Fig. 2D). The percentage of PD-L1 was significantly higher as compared with the uninfected control from 24 h post infection and peaked at 72 h (Fig. 2E). Overall, the data suggested that SFTSV infection induced the upregulation of PD-1/PD-L1 on T/B lymphocytes in a dynamic manner.

## PD-L1 upregulation in SFTSV infection of THP-1 cells

Since monocytes are the major cellular target for SFTSV infection (Zhang et al, 2019), human monocytic cell line THP-1 was infected with SFTSV (subtype E, JS-2013-14) at an MOI of 1, 3, or 10 for 48 h. As expected, the percentage of THP-1 cells expressing PD-L1 was proportional to the increasing viral dose, significantly upregulated at an MOI of 3 and 10, with nearly 50% cells expressing PD-L1 at an MOI of 10 (Fig. 3A). Correlational analysis showed that PD-L1 expression in THP-1 cells had strong positive correlation with viral RNA copies at different viral dose (Fig. 3B). To further determine the upregulation of PD-L1 in infected THP-1 over time course, we analyzed the cells by flow cytometry for phenotypic markers with SFTSV infection (MOI = 1) at various time points. As shown in Fig. 3C, the expression of PD-L1 significantly increased from 24 h post infection and continued to 120 h post infection when SFTSV replicated in THP-1 cells (Fig. EV4A), thus confirming the PD-L1 upregulation in SFTSV-infected THP-1 cells. Meanwhile, compared to the uninfected control, the expression of annexin V$^+$ of THP-1 cells with SFTSV infection was significantly upregulated from 24 h post infection and continued to 120 h post infection, indicating cell apoptosis in SFTSV-infected THP-1 cells (Fig. EV4B). The correlational analysis showed that PD-L1 expression in THP-1 strong positively correlated with viral RNA at various time points (Fig. 3D). Together, these data demonstrated that PD-L1 was upregulated in infected THP-1 cells over time course.

Earlier studies by others and our group showed that SFTSV infection drove infected THP-1 differentiation toward macrophage (Jin et al, 2012; Zhang et al, 2019). To determine whether SFTSV infection influences the PD-L1 expression on macrophages, THP-1 cells were first induced to differentiate into macrophages by PMA, followed by infection with increasing doses of SFTSV (MOI = 1/3/ 10) for 48 h. Consistently, macrophages were shown to express significantly higher PD-L1 with the higher dose of virus, with nearly 40% cells expressing PD-L1 at an MOI of 10 (Fig. 3E). The percentage of macrophages expressing PD-L1 exhibited a strong positive correlation with viral RNA copies (Fig. 3F). The percentage of PD-L1 expression in macrophages increased continuously, and was correlated significantly with the time of infection starting from 24 hpi (Fig. 3G), when SFTSV replicated in macrophages (Fig. EV4C). Figure 3H shows that the percentage of macrophages expressing PD-L1 of macrophages exhibited strong positive correlation with viral RNA in 12–120 h.

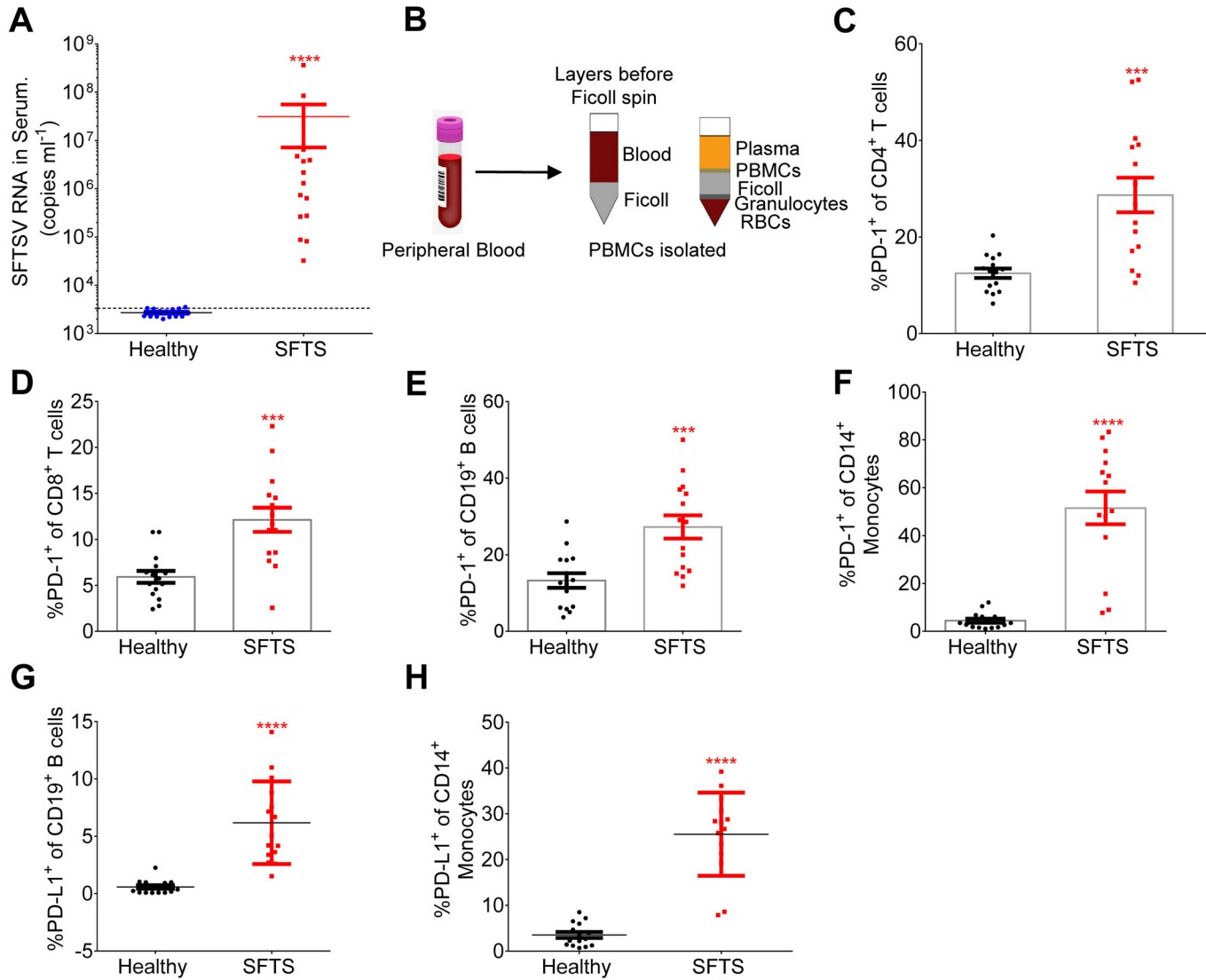

**Figure 1. PD-1/PD-L1 was upregulated in immune cells of SFTS patients.**

(A) Viral loads in serum among SFTS patients ($n = 15$) were measured by RT-PCR. Two-tailed unpaired $t$ test was performed to compare SFTS patients with healthy control (****$P < 0.0001$). (B) Peripheral blood mononuclear cells (PBMCs) were isolated using density-gradient centrifugation with Ficoll-Hypaque-gradient separation. (C–F) The expression of PD-1 in CD4+ (C) and CD8+ (D) T, CD19+ B cells (E), and CD14+ monocytes (F) was summarized for the SFTS patients ($n = 15$) and the healthy control ($n = 15$). Two-tailed unpaired $t$ test was performed to compare SFTS patients with healthy control (***$P = 0.0002$ (C, D); ***$P = 0.0005$ (E); ****$P < 0.0001$(F)). (G, H) The expression of PD-L1 in CD19+ B cells (G) and CD14+ monocytes (H) was summarized for the SFTS patients ($n = 15$) and the healthy control ($n = 15$). Two-tailed unpaired $t$ test was performed to compare SFTS patients with healthy control (****$P < 0.0001$). Data information: (A, C–H) data are shown as mean ± SEM. ***$P < 0.001$; ****$P < 0.0001$. Source data are available online for this figure.

## Characterization of the binding specificity of NbP45

To explore whether antibodies targeting PD-1 could inhibit SFTSV replication, we identified a new anti-PD-1 nanobody, named NbP45 that was isolated from an alpaca immunized with human PD-1. The purified NbP45 was verified by SDS-PAGE to ensure high purity (Fig. 4A). Tislelizumab (BGB-A317), an anti-human PD-1 monoclonal IgG4 antibody (Lee and Keam, 2020), served as a positive PD-1-blocking antibody control. Like Tislelizumab, NbP45 mainly reacted with non-reduced PD-1 protein in western blot (Fig. 4B), suggesting that NbP45 likely recognized a conformational epitope on PD-1. Next, the binding kinetics of NbP45 with PD-1

was tested by biolaye interferometry (BLI) (Table EV1), and NbP45 exhibited dose-dependent specific binding to PD-1 protein with a $K_D$ value of 8.64 nM at 300 nM (Fig. 4C). ELISA results showed that NbP45 or Tislelizumab displayed the similar binding curve with PD-1, with an $EC_{50}$ value of 12.94 ng/ml and 13.11 ng/ml, respectively (Fig. 4D). Flow cytometric analysis showed that NbP45 or Tislelizumab both could recognize PD-1 on cells, with $EC_{50}$ of 162.5 ng/ml and 94.63 ng/ml, respectively (Fig. 4E). Importantly, NbP45 or Tislelizumab blocked PD-1 binding to PD-L1 with an $IC_{50}$ of 1043 ng/ml and 779.3 ng/ml, respectively, as detected by FACS (Fig. 4F). Altogether, these results indicated that NbP45 recognized a conformational epitope on PD-1 with high

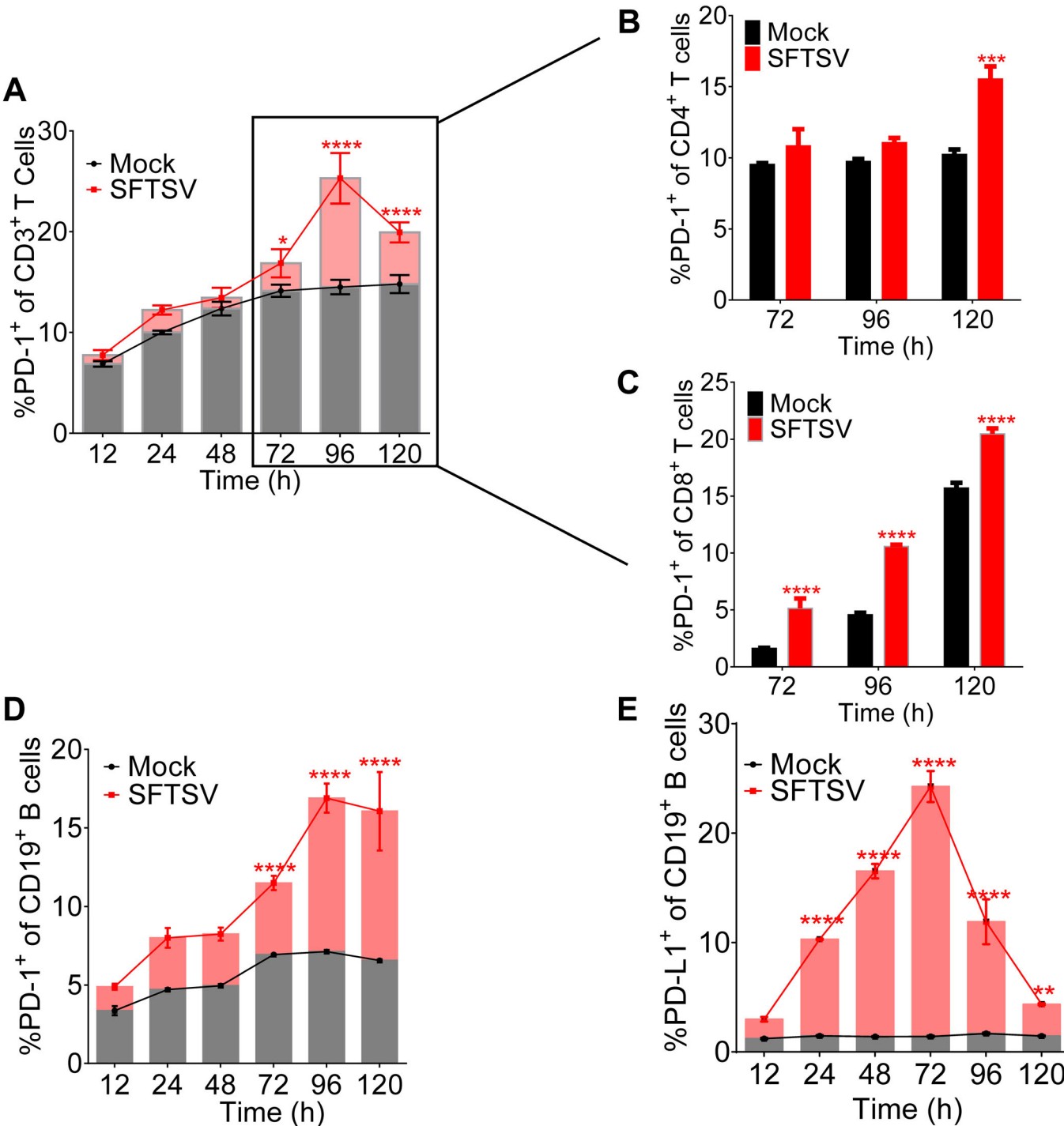

**Figure 2. SFTSV infection induced PD-1/PD-L1 upregulation on T/B lymphocytes.**

(A) The PD-1 expression of CD3+ T cells was summarized for the uninfected controls ($n = 3$) and the SFTSV (MOI = 1) infection ($n = 3$) at various time points. Two-way ANOVA with Tukey's test was performed to compare SFTSV infection with uninfected control (*$P = 0.0216$; ****$P < 0.0001$). (B, C) The PD-1 expression of CD4+ (B) and CD8+ (C) T cell was summarized for the uninfected controls ($n = 3$) and the SFTSV (MOI = 1) infection ($n = 3$) at 72/96/120 h. Two-way ANOVA with Tukey's test was performed to compare SFTSV infection with uninfected control (***$P = 0.0002$; ****$P < 0.0001$). (D, E) The PD-1 (D)/PD-L1 (E) expression of CD19+ B lymphocytes was summarized for the uninfected controls ($n = 3$) and the SFTSV (MOI = 1) infection ($n = 3$) at various time points. Two-way ANOVA with Tukey's test was performed to compare SFTSV infection with uninfected control (**$P = 0.0069$; ****$P < 0.0001$). Data information: (A–E) data are shown as mean ± SEM. *$P < 0.05$; **$P < 0.01$; ***$P < 0.001$; ****$P < 0.0001$. The $n$ means the number of sample repeats in one experiment at the same time. Source data are available online for this figure.

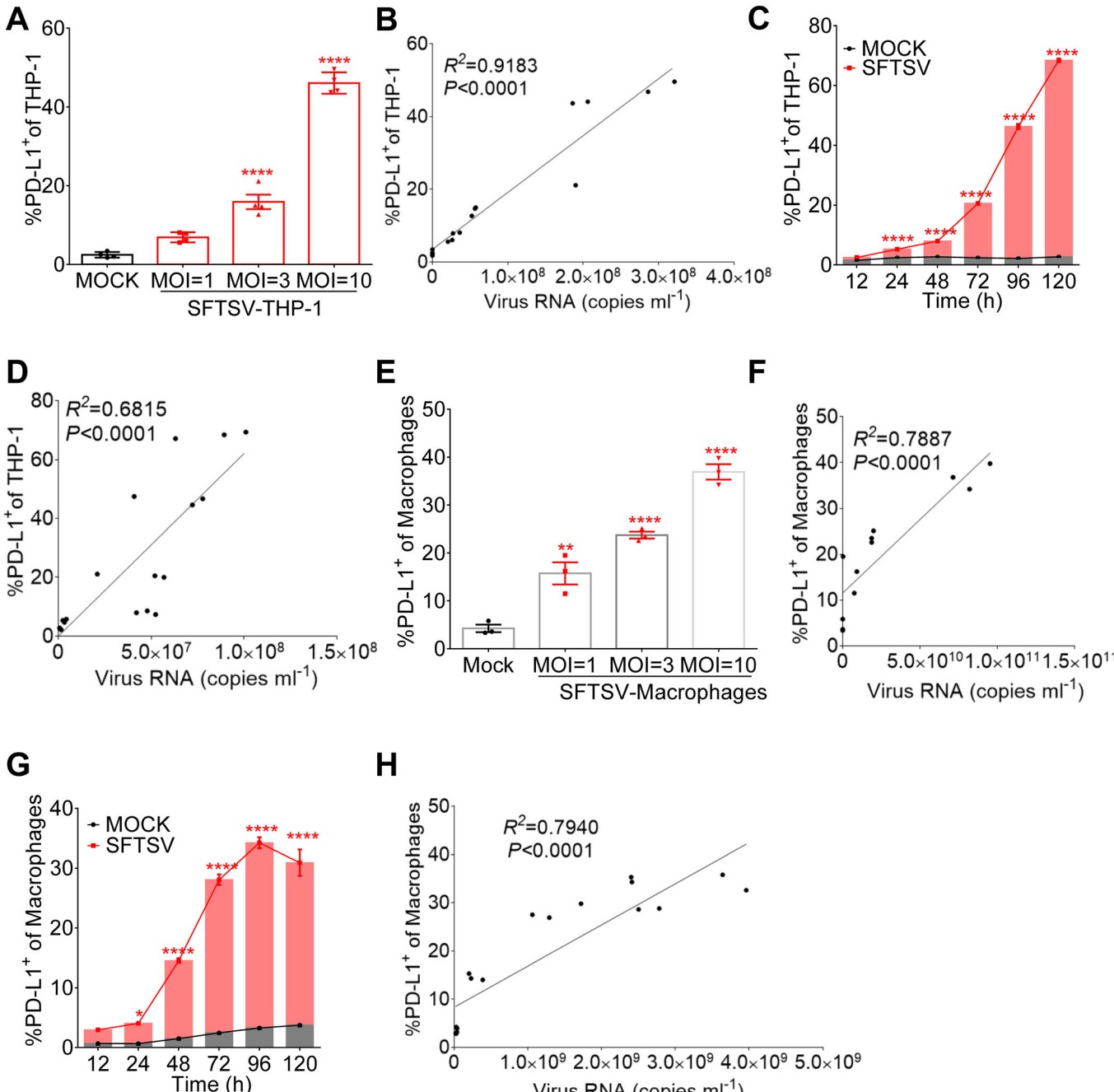

**Figure 3. PD-L1 upregulation in SFTSV infection of THP-1 cells.**

(A) The PD-L1 expression of THP-1 cells was summarized for the uninfected controls ($n = 3$) and the SFTSV infection at an MOI = 1/3/10 ($n = 3$) for 48 h. One-way ANOVA with Tukey's test was performed to compare SFTSV infection with uninfected control (****$P < 0.0001$). (B) Correlation between PD-L1 expression and viral RNA copies in SFTSV-infected THP-1 at an MOI = 1/3/10 ($n = 3$). (C) The PD-L1 expression of THP-1 cells was summarized for the uninfected controls ($n = 3$) and the SFTSV (MOI = 1) infection ($n = 3$) at various time points. Two-way ANOVA with Tukey's test was performed to compare SFTSV infection with uninfected control (****$P < 0.0001$). (D) Correlation between PD-L1 expression and viral RNA copies in SFTSV (MOI = 1) infected THP-1 at various time points. (E) The PD-L1 expression of macrophages was summarized for the uninfected controls ($n = 3$) and the SFTSV infection at an MOI = 1/3/10 ($n = 3$) for 48 h. One-way ANOVA with Tukey's test was performed to compare SFTSV infection with uninfected control (**$P = 0.0030$; ****$P < 0.0001$). (F) Correlation between PD-L1 expression and viral RNA copies in SFTSV-infected macrophages at an MOI = 1/3/10 ($n = 3$). (G) The PD-L1 expression of macrophages was summarized for the uninfected controls ($n = 3$) and the SFTSV (MOI = 1) infection ($n = 3$) at various time points. Two-way ANOVA with Tukey's test was performed to compare SFTSV infection with uninfected control (*$P = 0.0219$; ****$P < 0.0001$). (H) Correlation between PD-L1 expression and viral RNA copies in SFTSV (MOI = 1) infected macrophages at various time points. Data information: (A, C, E, G) data are shown as mean ± SEM. *$P < 0.05$; **$P < 0.01$; ****$P < 0.0001$. The $n$ means the number of sample repeats in one experiment at the same time. (B, D, F, H) Correlation analyses were performed by linear regression using the GraphPad Prism 6.0 program, Pearson's correlation tests were used to measure the strength of association between variables. Source data are available online for this figure.

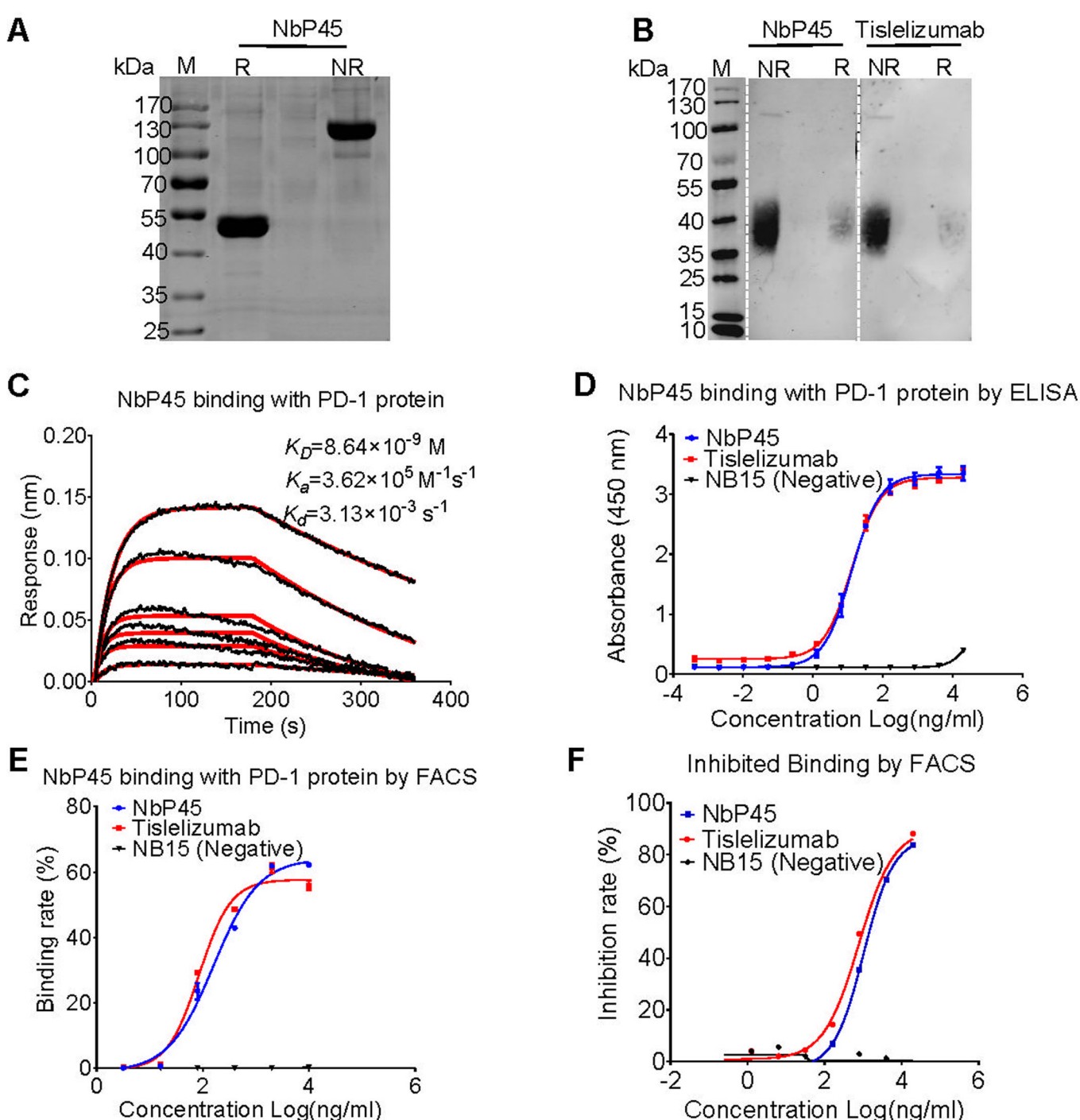

**Figure 4. Characterization of binding specificity of NbP45.**

(A) The purity of NbP45 was determined by SDS-PAGE under nonreducing (NR) or reducing condition (R). (B) PD-1 protein under nonreducing conditions (NR) or reducing conditions (R) was detected by western blot with NbP45 or Tislelizumab specific for PD-1 protein. (C) Kinetic binding curve of NbP45 at the concentration 300 nM, 100 nM, 33.3 nM, 11.1 nM, 3.7 nM and 1.2 nM with PD-1 proteins by BLI. (D, E) The binding curve of NbP45 ($n = 2$) or Tislelizumab ($n = 2$) with PD-1 protein detected by ELISA (D) and by FACS (E). The black line was taken as a negative control. (F) The Inhibition curve of NbP45 or Tislelizumab blocked PD-1 binding to PD-L1 was detected by FACS. Data information: (D, E) data are shown as mean ± SEM.

affinity, and blocked PD-1 binding to PD-L1 with an $IC_{50}$ value of 1043 ng/ml.

## NbP45 elicited potent attenuation of T-cell apoptosis, facilitated robust proliferation, and demonstrated broad-spectrum inhibition of SFTSV replication

To assess the impact of PD-1 blockade on SFTSV replication, PBMCs derived from healthy individuals were treated with NbP45 or Tislelizumab, followed by infection with SFTSV (subtype E, JS-2013-14) at an MOI of 1. Viral RNA quantification at 48 h post infection revealed significant inhibition of SFTSV replication by NbP45 or Tislelizumab compared to the untreated control, resulting in a reduction of viral RNA copies by more than $1.0 \times 10^7$ copies/ml (Fig. 5A). Intriguingly, the ratio of $CD8^+/CD4^+$ T cells was notably upregulated in the treated samples compared to the control (Fig. 5B), while the expression of annexin $V^+$ in $CD4^+/CD8^+$ T cells was notably downregulated (Fig. 5C,D). In addition, both $CD4^+$ (Fig. 5E–G) and $CD8^+$ T cells (Fig. 5H–J) treated with NbP45 or Tislelizumab exhibited significantly higher expression levels of HLA-DR, Ki-67, and IFN-γ, indicating the activation of T lymphocytes upon alleviating the PD-1/PD-L1 blockade. Furthermore, to evaluate the impact of PD-1 blockade on different subtypes of SFTSV replication, PBMCs from healthy individuals were treated with NbP45 or Tislelizumab, followed by infection with SFTSV (subtypes D and B) at an MOI of 1. Viral RNA quantification at 48 h post infection, compared to the untreated control, demonstrated significant inhibition of SFTSV replication (subtypes D and B) by NbP45 or Tislelizumab, indicating a broad-spectrum effect on different subtypes of SFTSV infection (Fig. EV5A,B). In summary, NbP45 or Tislelizumab demonstrated broad inhibition of SFTSV replication in PBMCs by attenuating immunological blockade, resulting in the downregulation of annexin V and the upregulation of HLA-DR, Ki-67, and IFN-γ in T lymphocytes.

## NbP45 administered via s.c. exhibited long-lasting in vivo kinetics

In light of the comparable inhibitory activity observed in vitro between NbP45 or Tislelizumab, a subsequent investigation was undertaken to evaluate their respective in vivo pharmacokinetic profiles in a mouse model. Following intraperitoneal administration (i.p.) of NbP45 or Tislelizumab, the total absorption of the drug within 336 h (Area Under Curve, AUC), maximum observed plasma concentration ($C_{max}$), and time of maximum observed plasma concentration ($T_{max}$) were determined. Specifically, in the serum of mice, the AUC values were measured as 23,885 and 16,372 µg/ml for NbP45 and Tislelizumab, respectively, while the corresponding $C_{max}$ values were 282.6 and 237.6 µg/ml. The $T_{max}$ values for NbP45 and Tislelizumab were found to be 4 and 8 h, respectively (Fig. 6A). Upon subcutaneous administration (s.c.) of NbP45 or Tislelizumab, the AUC, $C_{max}$, and $T_{max}$ were assessed as follows: 1489 and 711.1 µg/ml, 283.1 and 157.9 µg/ml, 72 and 48 h, respectively (Fig. 6B). These results indicate that the AUC of NbP45 was approximately twofold higher than that of subcutaneously administered Tislelizumab. To investigate the tissue distribution of the antibodies, mice were administered with YF®750 SE-labeled NbP45 or Tislelizumab (NbP45-YF750 or Tislelizumab-YF750) via

i.p. or s.c. analysis. Notably, in mice receiving subcutaneous NbP45, fluorescence signals were detected in the lung, liver, kidneys, and stomach after 336 h (14 days), with fluorescence levels reaching $\sim10^{11}$ ph/s (Fig. 6C,D). In addition, NbP45-YF750, when administered s.c., exhibited prolonged persistence in the liver, lung, and stomach for more than 336 h (14 days) (Fig. 6E). Compared to subcutaneously administered Tislelizumab, NbP45 displayed higher fluorescence signals, peaking at $1.5 \times 10^{13}$ ph/s at the site of subcutaneous injection after 4 h (Fig. 6F). Collectively, these findings underscore the superior antibody kinetics of subcutaneously administered NbP45 in vivo, establishing it as a promising route for therapeutic administration.

## Subcutaneous NbP45 was highly efficacious against SFTSV infection in humanized NCG mice

To evaluate the in vivo efficacy of NbP45, a specific model of humanized NCG mice with human peripheral blood lymphocytes (NCG-HuPBL) was employed, which allows for the assessment of human PD-1-blocking antibodies and can be infected with SFTSV. The NCG-HuPBL mice were challenged with $6 \times 10^5$ $TCID_{50}$ SFTSV (subtype E JS-2013-24) following established protocols (Li et al, 2021). Subsequently, the SFTSV-infected NCG-HuPBL mice ($n = 8$) were treated with 400 µg/mouse of NbP45 via intraperitoneal (i.p.) or subcutaneous (s.c.) administration at specific time points: 1, 3, 5, and 7 dpi (Fig. 7A). Control groups consisted of infected mice treated with PBS or NB15, a negative control nanobody, following the same treatment schedule. Notably, when NbP45 was administered subcutaneously, a significant inhibition of SFTSV replication was observed from day 9 to day 15. The experimental mice exhibited a reduction in viremia by over 2 $\log_{10}$ at 9 days post infection, followed by a reduction of 1 $\log_{10}$ at both 12 and 15 days post infection (Fig. 7B). Furthermore, the subcutaneous administration of NbP45 resulted in higher platelet counts compared to the control group, indicating its ability to prevent virus-induced thrombocytopenia (Fig. 7C). In line with the decrease in viral load, SFTSV-infected NCG-HuPBL mice treated with subcutaneous NbP45 demonstrated a significant increase in the ratio of $CD8^+/CD4^+$ T cells in the peripheral blood (Fig. 7D). Additionally, a dosing regimen of subcutaneous NbP45 administered every other day for a total of four doses led to a gradual increase in serum antibody concentration from 0 to 6 days post infection, surpassing the levels achieved with intraperitoneal administration. Afterward, the serum concentration decreased gradually after 8 days post infection (Fig. 7E). Importantly, correlation analysis revealed an inverse relationship between viral load and platelet counts (Fig. 7F). These collective findings demonstrate the effective inhibition of SFTSV replication and the alleviation of virus-induced thrombocytopenia through subcutaneous administration of NbP45 in the NCG-HuPBL mouse model.

## Discussion

In this study, we provided evidence that the SFTSV infection significantly upregulated the expression of PD-1/PD-L1 in T/B lymphocytes, monocytes and macrophages. NbP45, one of our new anti-PD-1 nanobodies which was isolated from an alpaca immunized with human PD-1, inhibited SFTSV infection of

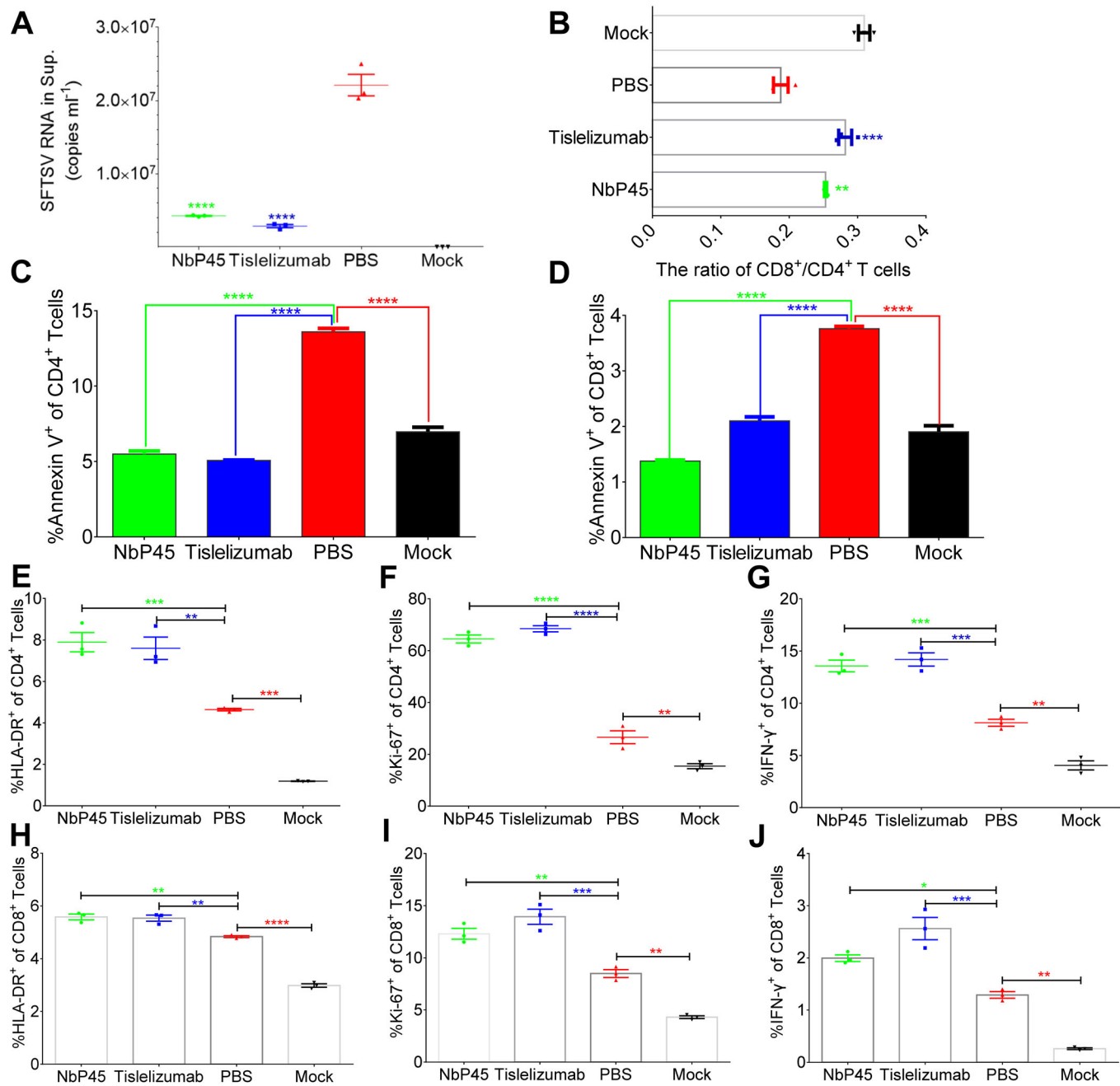

**Figure 5. NbP45 reduced apoptosis and enhanced proliferation of T lymphocytes, and potently inhibited SFTSV replication.**

(A) The inhibition activity of NbP45 ($n=3$) or Tislelizumab ($n=3$) against SFTSV (MOI $=1$) infection PBMCs at 48 hpi. One-way ANOVA with Tukey's test was performed to compare the treatment group with the control group (PBS) (****$P < 0.0001$). (B) The ratio of CD8$^+$/CD4$^+$ T cells in NbP45 ($n=3$) or Tislelizumab ($n=3$) against SFTSV (MOI $=1$) infection PBMCs at 48 hpi. One-way ANOVA with Tukey's test was performed to compare the treatment group with the control group (PBS) (**$P = 0.0024$; ***$P = 0.0002$). (C, D) Annexin V$^+$ expression of CD4$^+$ (C) and CD8$^+$ (D) T cells was summarized for the untreated controls ($n=3$) and the treatment ($n=3$) at 48 hpi. One-way ANOVA with Tukey's test was performed to compare the treatment group with the control group (PBS) (****$P < 0.0001$). (E–G) The HLA-DR (E), Ki-67 (F), and IFN-γ (G) expression of CD4$^+$ T cells were summarized for the untreated controls ($n=3$) and the treatment ($n=3$) at 48 hpi. One-way ANOVA with Tukey's test was performed to compare the treatment group with the control group (PBS) (***$P = 0.0009$ (NbP45 vs. PBS), **$P = 0.0017$ (Tislelizumab vs. PBS), ***$P = 0.0006$ (PBS vs. Mock) (E); **$P = 0.0060$, ****$P < 0.0001$ (F); ***$P = 0.0003$ (NbP45 vs. PBS), ***$P = 0.0001$ (Tislelizumab vs. PBS), **$P = 0.0021$ (PBS vs. Mock) (G)). (H–J) The HLA-DR (H), Ki-67 (I), and IFN-γ (J) expression of CD8$^+$ T cells were summarized for the untreated controls ($n=3$) and the treatment ($n=3$) at 48 hpi. One-way ANOVA with Tukey's test was performed to compare the treatment group with the control group (PBS) (**$P = 0.0013$ (NbP45 vs. PBS), **$P = 0.0020$ (Tislelizumab vs. PBS), ****$P < 0.0001$ (PBS vs. Mock) (H); **$P = 0.0026$ (NbP45 vs. PBS), ***$p = 0.0002$ (Tislelizumab vs. PBS), **$P = 0.0014$ (PBS vs. Mock) (I); *$P = 0.0112$ (NbP45 vs. PBS), ***$P = 0.0003$ (Tislelizumab vs. PBS), **$P = 0.0011$ (PBS vs. Mock) (J)). Data information: Data are shown as mean ± SEM. *$P < 0.05$; **$P < 0.01$; ***$P < 0.001$; ****$P < 0.0001$. The $n$ means the number of sample repeats in one experiment at the same time. Source data are available online for this figure.

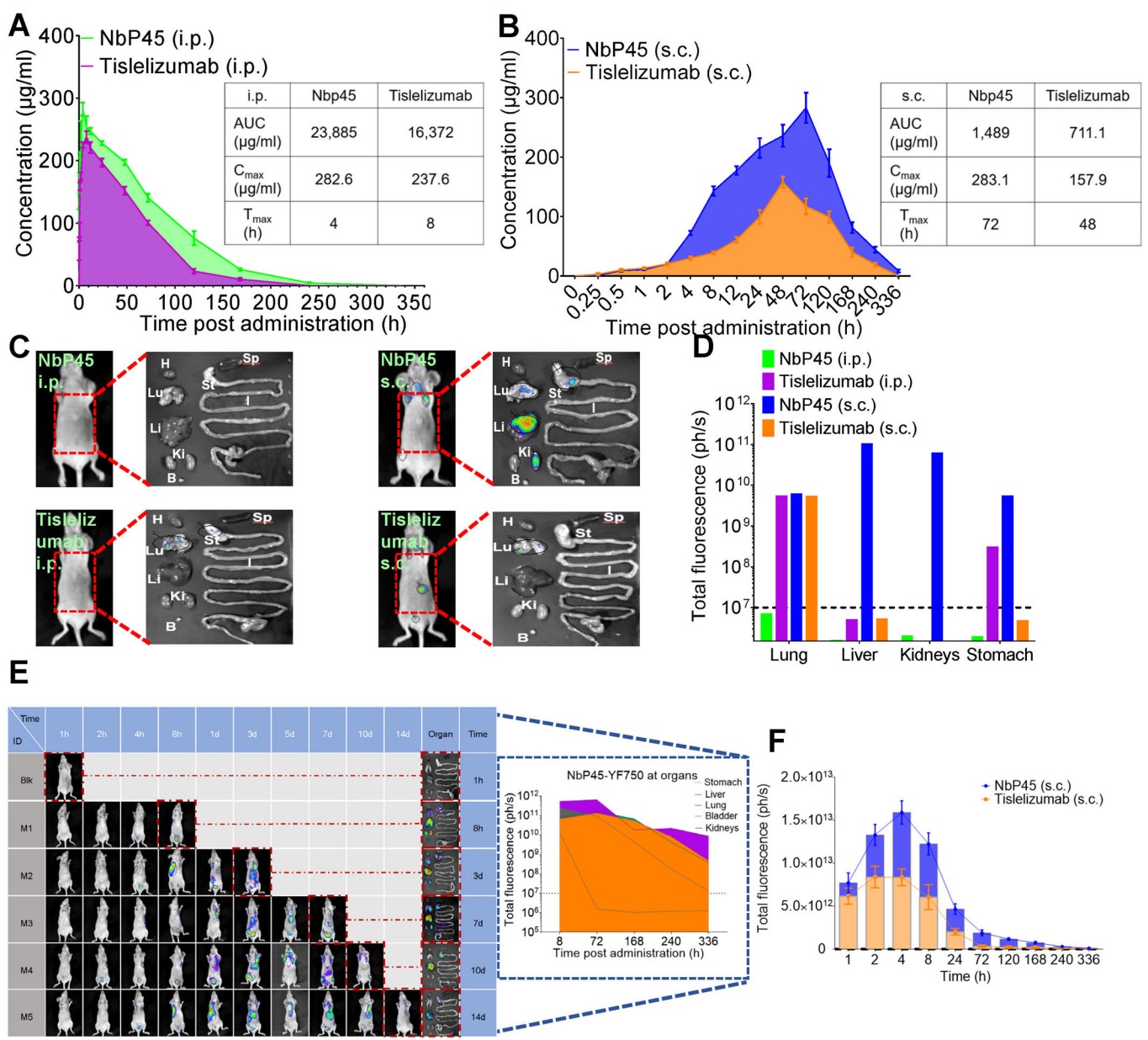

**Figure 6. NbP45 administrated via s.c. exhibited long-lasting in vivo kinetics.**

(A, B) Bioavailability of NbP45 or Tislelizumab in nude mice. NbP45 or Tislelizumab was administered into mice ($n = 4$, Female) at 400 μg/mouse via i.p. (A) or s.c. (B), respectively. Serum concentration of the NbP45 or Tislelizumab was determined at indicated time points by ELISA. AUC (area under the curve), the total absorption of the drug within 336 h after taking the drug, $C_{max}$, maximum observed plasma concentration, $T_{max}$, the time of maximum observed plasma concentration. (C) Spatial distribution of NbP45-YF750 or Tislelizumab-YF750 at 14 d after infusion into mice via i.p. or s.c. was detected by NightOwl LB 983. The right figure is the dissected image of the left mouse in the red dashed line. The right figure is the organs from dissected mouse which were imaged immediately after sacrifice. Lu lung, H heart, Li liver, Sp spleen, St stomach, I large and small intestine, Ki kidneys, B bladder. (D) The fluorescence intensity of organs in the organ column of (C) was summarized. Dashed line = limit of detection. (E) Spatial distribution of NbP45-YF750 via subcutaneous administration at indicated time point. Mice was sacrificed at the indicated time point for the analysis of fluorescence intensity in various organs. The figure of blue dashed line was the fluorescence intensity of organs for different time. (F) The fluorescence intensity of the site in subcutaneous injection of NbP45 ($n = 3$) or Tislelizumab ($n = 3$) was summarized. The black dashed line was the fluorescence intensity of non-injection control. Data information: (A, B, F) data represent as mean ± SEM. Source data are available online for this figure.

PBMCs through reducing apoptosis and enhancing the proliferation of T lymphocytes. Interestingly, compared to the licensed anti-PD-1 antibody drug, a conventional whole IgG, NbP45 injected subcutaneously (s.c.) exhibited better protective efficacy in SFTSV-infected humanized NCG mouse model (NCG-HuPBL).

The PD-1 pathway plays an important role in reducing the risk for autoimmunity and immunopathology (Sharpe and Pauken, 2018). PD-1 could downregulate T-cell activity during immune responses in order to prevent autoimmune tissue damage. PD-1-blocking antibodies have achieved great success in cancer

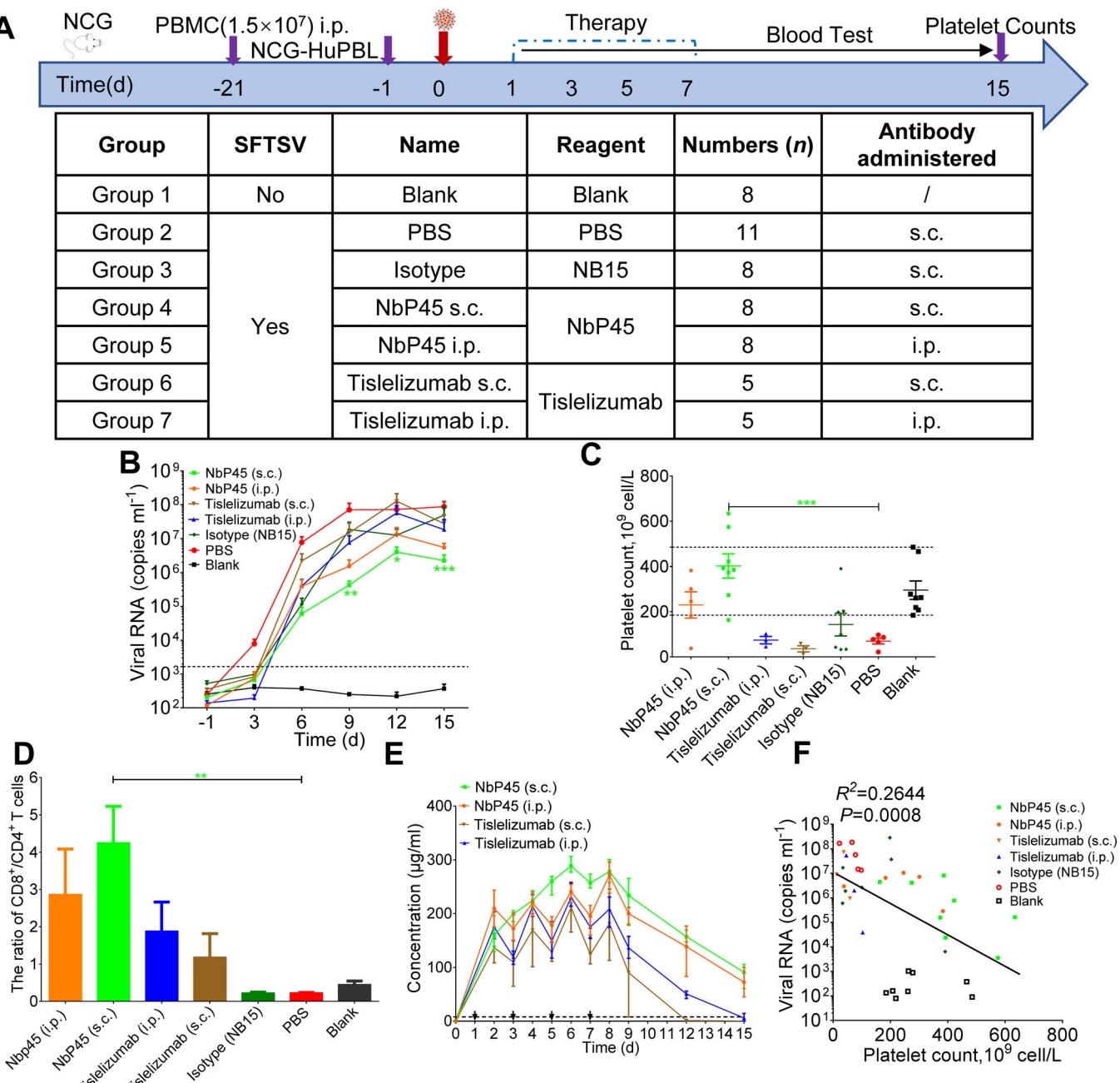

immunotherapy. In the chronic infection by numerous pathogens, lasting antigen exposure leads to permanent PD-1 expression that can cause the exhaustion of antigen-specific T cells (Blackburn et al, 2009), such as hepatitis B virus (HBV), hepatitis C virus (HCV), and human immunodeficiency virus (HIV), varicella zoster virus (VZV), and herpes simplex virus-1 (HSV-1), etc. (Channappanavar et al, 2012; Jeon et al, 2013; Jones et al, 2019; Jubel et al, 2020; Trautmann et al, 2006). The role of the PD-1 pathway during acute infection is less well-defined. In fact, there was evidence that its expression on dysfunctional CD4+ and CD8+ T cells was significantly increased, which may weaken the antiviral T-cell immunity against vaccinia virus, influenza virus, rabies in mice, and

hepatitis A virus (HAV), HBV, HCV, and EBOV (Jubel et al, 2020; Schönrich and Raftery, 2019).

SFTSV is an emerging infectious pathogen with high case fatality and increasing spatial distribution (Huang et al, 2021). Although the pathogenesis of SFTSV is not fully understood, increasing evidence suggests that SFTSV can evade host immune responses through interfering with functions of both innate and adaptive immunity (Jin et al, 2012; Li et al, 2021; Suzuki et al, 2020; Zhang et al, 2019). The adaptive immune response consists of T-cell-mediated cellular and B-cell-mediated humoral immunity. T lymphocytes play an important role in SFTSV antigen-specific immune responses (Li et al, 2021). The previous studies

**Figure 7. Subcutaneous NbP45 was highly efficacious against SFTSV infection of humanized NCG Mice.**

(A) Experimental schedule of NbP45 in the treatment of SFTSV infection. Bottom, table summary of groups with different treatments. (B) Plasma viral loads among seven groups of color-coded NCG-HuPBL mice including no SFTSV challenge ($n = 8$), SFTSV challenge with PBS treatment as control ($n = 11$), SFTSV challenge with isotype (NB15) treatment as control ($n = 8$), NbP45 treatment by s.c. ($n = 8$) or i.p. ($n = 8$), Tislelizumab treatment by s.c. ($n = 5$) or i.p. ($n = 5$). Each line represents data from 1 group. The limit of detection was 1000 genome copies/ml. Dashed line = limit of detection. Two-way ANOVA with Tukey's test was performed to compare the treatment group with the control group (PBS) (**$P = 0.0023$ (9 days); *$P = 0.0105$ (12 days); ***$P = 0.0009$ (15 days)). (C) Platelet count was calculated among the same color-coded groups of animals at 15 days, including no SFTSV challenge ($n = 8$), SFTSV challenge with PBS treatment as control ($n = 5$), SFTSV challenge with isotype (NB15) treatment as control ($n = 7$), NbP45 treatment by s.c. ($n = 8$) or i.p. ($n = 5$), Tislelizumab treatment by s.c. ($n = 3$) or i.p. ($n = 3$). The normal range is between the two dashed lines. Each dot represents data from 1 mouse. The blood from infected mice was collected at 15 days and hematological examination was performed using a hematology analyzer. One-way ANOVA with Tukey's test was performed to compare the treatment group with control group (PBS) (***$P = 0.0003$). (D) The ratio of human CD8$^+$/CD4$^+$ T cells from lymphocyte gate among seven groups of color-coded NCG-HuPBL mice at 15 days including no SFTSV challenge ($n = 8$), SFTSV challenge with PBS treatment as control ($n = 5$), SFTSV challenge with isotype (NB15) treatment as control ($n = 7$), NbP45 treatment by s.c. ($n = 8$) or i.p. ($n = 5$), Tislelizumab treatment by s.c. ($n = 3$) or i.p. ($n = 3$). One-way ANOVA with Tukey's test was performed to compare the treatment group with control group (PBS) (**$P = 0.0032$). (E) Serum concentration of the NbP45/Tislelizumab was determined at indicated time points by ELISA. (F) The correlation between plasma viral load and platelet counts among the same color-coded groups of animals. Correlation analyses were performed by linear regression using the GraphPad Prism 6.0 program, Pearson's correlation tests were used to measure the strength of association between variables. Data information: (B–E) data represent mean ± SEM. *$P < 0.05$; **$P < 0.01$; ***$P < 0.001$. Source data are available online for this figure.

demonstrated that CD3$^+$, CD4$^+$, and CD8$^+$ T cells were diminished in SFTS patients (Sun et al, 2014; Yi et al, 2015) and in experimental animal models (Jin et al, 2012). In this study, we presented evidence that the peripheral CD4$^+$ and CD8$^+$ T cells in the SFTSV infection were decreased (Fig. EV3A,B), consistent with a previous study by Sun et al (Sun et al, 2014) and Li et al (Li et al, 2018) that reported a significant decline of CD3$^+$, CD4$^+$, and CD8$^+$ expression in the peripheral blood of SFTS patients. Meanwhile, SFTSV infection significantly induced PD-1 expression of T lymphocytes (Figs. 1C,D, 2A–C, and EV1A,B). B lymphocytes, particularly plasma cells, serving an integral role in the humoral immunity against viral infection, are potential virus reservoirs in the patients with fatal SFSTV infections (Park et al, 2021; Schultz et al, 2016). Our studies reported that the percentage of PD-1$^+$/PD-L1$^+$ CD19$^+$ B cells was significantly higher in the SFTSV-infected patients (Fig. 1E–G) and the expression level of PD-1$^+$/PD-L1$^+$ in the B cells was also elevated (Fig. EV1C–E). By contrast, a previous patient cohort study found that the number of B cells was markedly reduced during the first week, but quickly returned to normal levels in SFTS patients (Lu et al, 2015). Our data showed that the expression of PD-1 increased gradually from 12 h post infection and reached a significantly higher level after 72 h (Fig. 2D), and PD-L1 expression was significantly higher compared with the uninfected control from 24 h post infection and peaked at 72 h (Fig. 2E). Our and other previous studies demonstrated that both NP- and Gn-specific IgGs were absent from the deceased patients, and the absence of the Gn-specific neutralizing antibody was strongly correlated with the death (Lu et al, 2015; Song et al, 2018). Thus, modulating dysfunctional immune cells can be a potential strategy to restore host immune clearance against SFTSV (Park et al, 2021).

Inhibition of the PD-1 pathway has a profound impact on tumor biology and cancer immunotherapy. So far, several monoclonal antibodies, blocking the interaction of PD-1 and PD-L1, such as Nivolumab, Pembrolizumab, and Tislelizumab, et al can reverse immunosuppressive conditions and improve the killing of tumor cells by host's immune cells (tumor-specific CD8$^+$ T cells) (Han et al, 2020), and are used to treat advanced melanoma, non-small cell lung cancer, and head and neck squamous cell cancer (Kraehenbuehl et al, 2022). In humanized BLT mice infected with HIV, treatment with PD-1 antibody increased CD4$^+$ T-cell levels, coordinated with reduced HIV-1 viral loads (Palmer et al, 2013). In

other studies, blocking PD-1 could enhance CD8$^+$ T-cell function and improve CD4$^+$ T-cell reconstitution in the gut of simian immunodeficiency virus (SIV)-infected rhesus macaques, resulting in a significant delay in viral rebound and a reduction of the viral setpoint following ART interruption (Bekerman et al, 2019; Muthumani et al, 2008; Pan et al, 2018). Though PD-1 upregulation and its immune inhibitory roles were observed in acute infections such as EBOV, SFTSV, antiviral therapies targeting PD-1 have not been reported. In the current study, NbP45 or Tislelizumab demonstrated broad inhibited the SFTSV infection of PBMCs (Figs. 5A and EV5), with the expression of Annexin V$^+$ being downgraded, HLA-DR, Ki-67 and IFN-γ of CD4$^+$/CD8$^+$ T cells significantly elevated (Fig. 5E–J). Apoptosis is a strictly regulated physiological process that is closely related to innate immunity. A previous study found that the expression levels of apoptosis markers (annexin V) on CD4$^+$ and CD8$^+$ T cells were significantly higher in SFTS patients (Li et al, 2018). When PD-1 was blocked by NbP45 or Tislelizumab, the expression of Annexin V$^+$ of CD4$^+$/CD8$^+$ T cells was significantly downregulated, enhanced T-cell immune response and inhibited SFTSV-infected PBMCs (Fig. 5). In addition, compared to the licensed anti-PD-1 antibody drug with traditional format of whole IgG, our data suggested that subcutaneous NbP45 had better antibody kinetics in vivo (Fig. 6), and exhibited better protective efficacy in SFTSV-infected humanized NCG mouse model (NCG-HuPBL) (Fig. 7). These results suggest that PD-1/PD-L1 pathway is involved in the acute infection caused by SFTSV, and could serve as the conserved host targets for developing potential immunotherapy interventions to broadly treat SFTSV infection.

Various therapeutic approaches have been attempted to treat SFTS, such as antiviral agents, steroid therapy, intravenous immunoglobulin (IVIG), plasma exchange, and monoclonal antibodies (Choi et al, 2018; Jung et al, 2021; Kim et al, 2019; Shimojima et al, 2022; Wu et al, 2020). RNA viruses are characterized by high mutation rates and a high potential for recombination, leading to a high genomic heterogeneity, which often leads to fast emerging of resistant mutants (Moya et al, 2004). Remarkably, a distinct geographical distribution pattern emerges among SFTSV strains, particularly evident in the majority of those originating from mainland China, which predominantly align with genotypes B and D. In contrast, strains isolated from South Korea, Japan, and Zhoushan Islands of Zhejiang Province, China, exhibit a

prevalence of genotype E according to phylogeographic analyses (Li et al, 2017a). This observation implies a plausible scenario of cross-ocean transmission for the genotype E virus, underscoring the intricate dynamics of SFTSV dissemination across diverse regions. (Hu et al, 2023; Li et al, 2016; Li et al, 2017a; Yun et al, 2017; Yun et al, 2020; Zhan et al, 2017). Immune checkpoint proteins, such as PD-1, PD-L1, CTLA-4, CD47, and TIGIT, play crucial roles in preventing immune-driven pathology. However, their regulatory functions may also restrict the immune-mediated clearance of infections. Immunotherapy, designed to enhance the host immune response against disease (Rowshanravan et al, 2018; Wykes and Lewin, 2018), suggests that targeting immune checkpoint proteins holds promise as potential therapeutic intervention for addressing SFTSV infection. In our current study, we have identified the role of PD-1, while the involvement of other immune checkpoint proteins, including PD-L1, CTLA-4, CD47, TIGIT, among others, requires further investigation. Taken together, the treatment using antibodies against immune checkpoints has several unique advantages: (1) although SFTSV has 6 different genotypes and multiple different serotypes, treatments targeting immune checkpoints will not be easily affected by differing genotypes; (2) targeting immune checkpoints can overcome the intrinsic nature of highly mutated RNA viruses; (3) more effective inhibition and clearance of viruses may be achieved by activating the immune system.

# Methods

## Patients and clinical samples

Clinical specimens were obtained from 15 SFTS patients admitted into Nanjing Drum Tower Hospital from March to July in 2022. The patients were confirmed of SFTSV infection by RT-PCR. Whole blood was collected by venipuncture in Vacutainer tubes containing EDTA (ethylenediaminetetraacetic acid) (Becton Dickinson, Rutherford, NJ). Peripheral blood mononuclear cells (PBMCs) were isolated by Ficoll-Hypaque gradient separation and washed twice in phosphate-buffered saline (PBS; Organon Teknica), and the number of viable leukocytes was determined by trypan blue exclusion. All analyses were performed on freshly collected cell. Serum specimens were stored at −80 °C until used. Informed consent was obtained from all subjects, in accordance with the Declaration of Helsinki and principles set out in the Department of Health and Human Services Belmont Report, and the research was approved by the Ethics Committee of Nanjing Drum Tower Hospital.

## PBMCs and cell lines

Peripheral blood mononuclear cells (PBMCs) (derived from healthy donors visiting the Drum Tower Hospital, Nanjing University) were isolated following our previously published protocol (Xu et al, 2021b), and cultured in RPMI-1640 (Gibco, USA), 10% heat-inactivated FBS and freshly added recombinant human interleukin-2 (IL-2) (20 ng/ml). Human monocyte cell line (THP-1) was purchased from Cell Resource Center of Shanghai Institute for Biological Sciences, Chinese Academy of Sciences, China. THP-1 cells were differentiated in RPMI-1640 (Gibco, USA) medium supplemented with 10% FBS containing 5 mM L-glutamine, penicillin/streptomycin, and Phorbol-12-myristate-13-acetate (PMA 50 nM, Sigma, CA) for 24 h followed by culturing in fresh medium for another 24 h. 293TT cells (ATCC) were cultured in Dulbecco's Modified Eagle Medium (DMEM) (Thermo Fisher Scientific) containing 10% Fetal bovine serum, 100 U/mL penicillin and 2 mM L-glutamine and were incubated at 37 °C in 5% $CO_2$ setting. 293-F cells (Thermo Fisher Scientific) were cultured in Expi293TM Expression Medium and were incubated in a 37 °C incubator and 8% $CO_2$ setting on an orbital shaker platform at 130 rpm according to the manufacturer's instructions.

## Viral load determination by quantitative real-time PCR

The viral RNA in cellular supernatant, cells, patients and mouse' serum were extracted using Quick-RNA Viral Kit (ZYMO Research, CA, USA) or TRIzol reagent (Thermo Fisher Scientific, MA, USA) and cDNAs were generated at 37 °C for 15 min, 85 °C for 5 s, and 37 °C for 10 min using RT-PCR Prime Script Kit (Takara, CA, USA). Quantitative real-time PCR analyses for mRNAs of SFTSV was performed by using TaqMan® Gene Expression Assays (Thermo Fisher, USA) and ABI 7500 Real-time PCR system (Life Tech, USA) according to the manufacturer's procedure (Yu et al, 2011). The primer set was as below: forward primer: 5′-TGGTGGATGTCATAGAGG G-3′, reverse primer: 5′-CTGTG TTCACTGTTGATTTCTC-3′, probe: 5′-AGCATACGCCCTAAG TCAGACATGGATGA-3′. Viral copy numbers were calculated as a ratio with respect to the standard control (Shimada et al, 2015). Gene expression of relative fold change, recorded as cycle threshold (Ct), was normalized against an internal control (GAPDH).

## Flow cytometric analysis

All antibodies for flow cytometry were purchased from BioLegend or Invitrogen. PBMCs and THP-1 cells were resuspended in ice-cold flow cytometry buffer (2% [v/v] FBS and 2 mM EDTA in PBS). After nonspecific binding being blocked by Human TruStain FcX™ (Fc Receptor Blocking Solution) (BioLegend, USA), cells were stained with antibodies (according to the experiment's requirement), and the antibodies were grouped into seven panels for the phenotypic analysis of SFTS PBMCs (A/B), 293TT-PD-1/PD-L1 (C), THP-1 and macrophages (D), T-cell proliferation and activation (E), T-cell apoptotic (F), Humanized NCG mouse (G). Red blood cells were removed by lysing in Red Blood Cell Lysis Buffer (BD). Antibody combination in each panel was as follow: Panel A: FITC anti-human CD3, APC-Cy7 CD4 anti-human, PE anti-human CD8, PE/Cyanine 7 anti-human CD279 (PD-1), APC anti-human CD274 (B7-H1,PD-L1); Panel B: Pacific Blue™ anti-human CD19, PE anti-human CD14 antibody, PE/Cyanine 7 anti-human CD279 (PD-1), APC anti-human CD274 (B7-H1,PD-L1); Panel C: PE/Cyanine 7 anti-huamn CD279 (PD-1), APC anti-human CD274 (B7-H1,PD-L1); Panel D: APC anti-human CD274 (B7-H1,PD-L1); Panel E: FITC anti-human CD3, PE-Cy7 mouse anti-human CD4, BV510 mouse anti-human CD8 (SK1), BV421 mouse anti-human HLA-DR (G46-6), PE mouse anti-Ki-67, APC mouse anti-human IFN-γ (B27); Panel F: APC-Cy7 anti-human CD3, PE-Cy7 mouse anti-human CD4, BV510 mouse anti-human CD8 (SK1), Annexin V-FITC/PI apoptosis detection kit (Catalog #40302ES5); Panel G: APC anti-human CD45, FITC mouse anti-human CD3, PE-Cy7 mouse anti-human CD4, PE mouse anti-human CD8. To detect intracellular expression of Ki-67 and IFN-γ, PBMCs were fixed and permeabilized with BD Cytofix/Cytoperm kit (BD Biosciences). The cells were incubated with corresponding

target antibodies for 30 min at 4 °C, then were washed twice and suspended in FACs buffer. All cells were measured and sorted by using flow cytometer (NovoCyte Flow Cytometer, ACEA) and analyzed with FlowJo software version 10.2.

## Expression and purification of NbP45

The Fc gene of the human monoclonal antibody was fused with the VHH gene of nanobodies (named as Nb-Fc or Nbs) to assist the purification and prolong the half-life of the Nb antibody, following our previously published protocol (Wu et al, 2020). The Nbs were finally cloned into an expression vector (Invitrogen), which were transfected into 293-F cells (cat# R79007, Thermo Scientific) to produce Nb-Fcs. Cells were harvested by centrifugation at 4500 rpm for 15 min, resuspended and homogenized in the lysis buffer containing 20 mM Tris–HCl, 150 mM NaCl, pH 7.5 using ultrasonic. Cell debris was removed by centrifugation at 18,000 rpm for 30 min. The supernatants were added to Ni-NTA resin (GE Healthcare, USA). The nonspecific contaminants were eluted by washing the resin with the lysis buffer containing 10 mM imidazole. The nanobody was subsequently eluted with the lysis buffer containing 500 mM imidazole. NbP45 was eluted and purified by Superdex 75 (GE Healthcare, USA).

## SDS-PAGE and western blotting

Purified protein or antibody was separated under nonreducing (NR) or reducing condition (R) by electrophoresis in a 10% SDS-PAGE gels, and transferred onto a 0.22-μm PVDF membrane. The membrane was soaked in blocking buffer (2% BSA in PBST) at room temperature for 1 h, and then incubated the first blocked and then incubated at 4 °C overnight or 37 °C for 1 h, followed by a secondary antibody of anti-human IgG with an IRDye 800CW (926-32232, Rockland). Protein bands were visualized using Odyssey CLx Imaging System (LI-COR).

## Biolayer interferometry (BLI)

The affinity of NbP45-specific antibodies was determined using the ForteBio OctetRED 96 biolayer interferometry instrument (Molecular Devices ForteBio LLC, Fremont, CA), with human Fc tag, anti-human Fc (AHC) biosensor (cat# 18-5060, Fortebio) was used to immobilize the PD-1 protein. AR2G biosensors were activated for 7 min with EDC (400 mM)/NHS (100 mM), and then immersed 15 min in PD-1 protein which was diluted in pH 5.0 10 mM sodium acetate buffer. The kinetics assays were performed with a shaking speed of 1000 rpm. Association of Nbs in a serial dilution (300 nM, 100 nM, 33.3 nM, 11.1 nM, 3.7 nM, and 1.2 nM) was performed prior to dissociation for 180 s. The affinity analysis was performed using a fast 1:1 binding model and the Data analysis software 8.0 (Sartorius). $K_D$, $K_a$, and $K_d$ values were evaluated with a global fit applied to all data.

## ELISA analysis

Antibody quantification, antibody characterization, and serum from mice were quantified by ELISA as our previously reported method (Wu et al, 2019). In brief, the protein was coated to ELISA plates (Corning) at a concentration of 0.5 mg/ml. After washing 2–4 times, 5% non-fat milk in PBS was added and incubated for

blocking at 37 °C for 1 h. After washing, 100 μl serially diluted sera (1:5000/25,000/125,000) or purified antibody (20, 4, 0.8, 0.16, 0.032, 0.0064, 0.00128, 0.000256, 0.0000512, 0.00001024, 0.000002048, 0.00000041 μg/ml) was added and incubated at 37 °C for 1 h. Following washing, secondary antibody of goat anti-human IgG with HRP (Jackson ImmunoResearch, 1:10,000 dilution) was added and incubated at 37 °C for 1 h. Accordingly, 3,3′,5,5′-Tetramethyl-benzidine (TMB, Sigma) substrate was added at 37 °C for 10 min; and 10 ml 0.2 M $H_2SO_4$ was added to stop the reaction. The optical densities at 450 nm (OD = 450) was measured using the Infinite 200 (Tecan, Ramsey, MN, USA). Antibody quantification in the sera was calculated according to the standard curve generated by purified antibody. Replicate of each sample was measured and three independent experiments were performed.

## PD-1 blocked against SFTSV live virus

In brief, NbP45 (20 μg/ml) or Tislelizumab (20 μg/ml) with $5.0 \times 10^5$ PBMCs for 1 h in 24-well plates at 5% $CO_2$, 37 °C. SFTSV (MOI = 1) of JS-2013-24 (subtype E) (GenBank accession no. AMY99382) (Li et al, 2017b) infection at 48 hpi. The viral RNA was determined by quantitative reverse transcription PCR (RT-PCR). In addition, ratio of $CD8^+/CD4^+$ T cells, annexin V (apoptotic markers), and HLA-DR/Ki-67/IFN-γ (T-cell proliferation, activation, and function markers) in T lymphocytes were measured by Flow cytometric analysis, respectively.

For another strain of SFTSV, NbP45 (20 μg/ml) or Tislelizumab (20 μg/ml) with $5.0 \times 10^5$ PBMCs for 1 h in 24-well plates at 5% $CO_2$, 37 °C. SFTSV (MOI = 1) of 20ZJTZ07 (subtype D) (GenBank accession no. UNP61901) or HB29 (Subtype B) (GenBank accession no. NC_018137) infection at 48 hpi. The viral RNA was determined by quantitative reverse transcription PCR (RT-PCR).

## Pharmacokinetics (PK) of Nbs in vivo

Purified Nbs were injected intraperitoneally (i.p.), subcutaneously (s.c.) into nude mice (Qing Long Shan Animal Center, Nanjing, China) at a dose of 20 mg/kg. ELISA was used to measure the serum concentration of Nbs. The AUC (Area Under Curve), $C_{max}$ (maximum observed plasma concentration) and $T_{max}$ (the time of maximum observed plasma concentration) were analyzed by the GraphPad software.

## Spatial distribution of Nb in vivo

Spatial distribution of Nb was conducted following our previously published protocol (Wu et al, 2022), with minor modifications. Nb was labeled with far infrared dye YF®750 SE (US EVERBRIGHT INC, YS0056) (named as Nb-YF750). Purified Nb-YF750 was injected s.c. or i.p. into nude mice (Qing Long Shan Animal Center, Nanjing, China) at a dose of 20 mg/kg. Images were observed at Ex:740 nm/Em:780 nm by NightOWL LB 983 (Berthold, Germany) at the indicated time point. Images were analyzed using Indigo imaging software Ver. A 01.19.01.

## Evaluating the efficacy of NbP45 in SFTSV-infected NCG-HuPBL mice

Immunodeficient NCG mice was purchased from GemPharmatech Co., Ltd., Nanjing, China. Similar to NOD.Cg-Prkd$^{cscid}$ Il2rg$^{tm1Wjl}$/

## The paper explained

### Problem

Severe fever with thrombocytopenia syndrome (SFTS) is a life-threatening disease caused by infection with a novel bunyavirus (SFTSV), which is mainly transmitted by tick bites, though human-to-human transmission has also been reported recently. SFTSV has shown increasing spatial distribution in recent years in Asia, and infected cases have been on the rise. The case fatality rate of SFTS is 6–30%, and no effective therapies or vaccines for SFTS are available.

### Results

We observed and explored the upregulation of PD-1/PD-L1 in clinical samples from SFTS patients. And revealing elevated PD-1/PD-L1 expression in various immune cells following SFTSV infection in vitro experiments.

Our findings demonstrate that NbP45, one of our new anti-PD-1 nanobodies isolated from an alpaca immunized with human PD-1, can inhibit SFTSV infection both in vitro and in an SFTSV-infected NCG-HuPBL mouse model. Interestingly, we also found that the subcutaneous administration of NbP45 nanobody exhibited better protective efficacy than the anti-PD-1 antibody drug with the traditional format of whole IgG, indicating its potential in overcoming SFTSV mutation.

### Impact

In conclusion, these findings highlight the involvement of the PD-1/PD-L1 pathway during the acute infection caused by SFTSV, and could serve as the conserved host targets for developing potential immunotherapy interventions to broadly treat SFTSV infection.

SZJ mice, the NCG mice lacked the IL-2 receptor gene in a SCID background, resulting in the absence of murine T and B cells and very small numbers of NK cells (Yu et al, 2017). In all, $2 \times 10^7$ human PBMCs were injected intraperitoneally (i.p.) into each of 4- to 6-week-old NCG mice. PBMCs engraftments were evaluated at the 21st day after the transplantation by FACS. Next, humanized NCG mice were challenged with $6 \times 10^5$ TCID$_{50}$ SFTSV (SFTSV E-JS-2013-24 strain) and conducted as we previously reported (Li et al, 2021). The infected NCG-HuPBL mouse were treated with NbP45 or Tislelizumab at 400 μg/mouse was administered via i.p. or s.c. 1, 3, 5, and 7 days post infection. Infected mice treated with PBS, NB15 at the same time intervals served as controls. At 0, 3, 6, 9, 12, and 15 days, the blood was collected for monitoring viral load by quantitative PCR, for monitoring the ratio of CD8$^+$/CD4$^+$ T cells by FACS and for completing blood count conducted at 15 days at Nanjing Drum Tower Hospital (Jiangsu, China). The animal study was approved by the Institutional Animal Welfare Committee and conducted in compliance with biosafety guidelines.

### Statistics

All statistical analyses were carried out using GraphPad Prism 8.01 software (GraphPad) or OriginPro 8.5 software (OriginLab). At least two independent biological replicates were performed for each experimental condition, of which details can be found in respective figure legends. Mice were randomized on the day of the experiment. Mice were allocated to cages so that each cage contained individuals representing each treatment. Two-tailed unpaired *t* test or ANOVA was performed for group comparisons.

Pearson's correlation tests were used to measure the strength of association between variables. *$P < 0.05$ was considered as statistically significant with mean ± SEM.

## Data availability

Figure 4A–F from the main figures are available at BioImage Archive. Accession no. S-BSST1245.

## Peer review information

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

## Acknowledgements

The study and the protocol for this research were approved by the Center for Public Health Research, Medical School, Nanjing University. The animal study was reviewed and approved by the Committee on the Use of Live Animals by the Ethics Committee of Nanjing University. This work was supported by National Science Foundation of China (NSFC) (No. 32370988, 31970149, and U22A20335), The Major Research and Development Project (2018ZX10301406), and Nanjing University-Ningxia University Collaborative Project (Grant# 2017BN04).

## Author contributions

**Mengmeng Ji**: Formal analysis; Visualization; Writing—original draft. **Jiaqian Hu**: Formal analysis; Visualization; Methodology. **Doudou Zhang**: Formal analysis. **Bilian Huang**: Formal analysis. **Shijie Xu**: Formal analysis; Methodology. **Na Jiang**: Methodology. **Yuxin Chen**: Resources; Methodology. **Yujiong Wang**: Resources; Methodology. **Xilin wu**: Resources; Data curation; Funding acquisition; Methodology; Writing—review and editing. **Zhiwei Wu**: Data curation; Supervision; Funding acquisition; Methodology; Writing—review and editing.

## Disclosure and competing interests statement

The authors declare no competing interests.

# Expanded View Figures

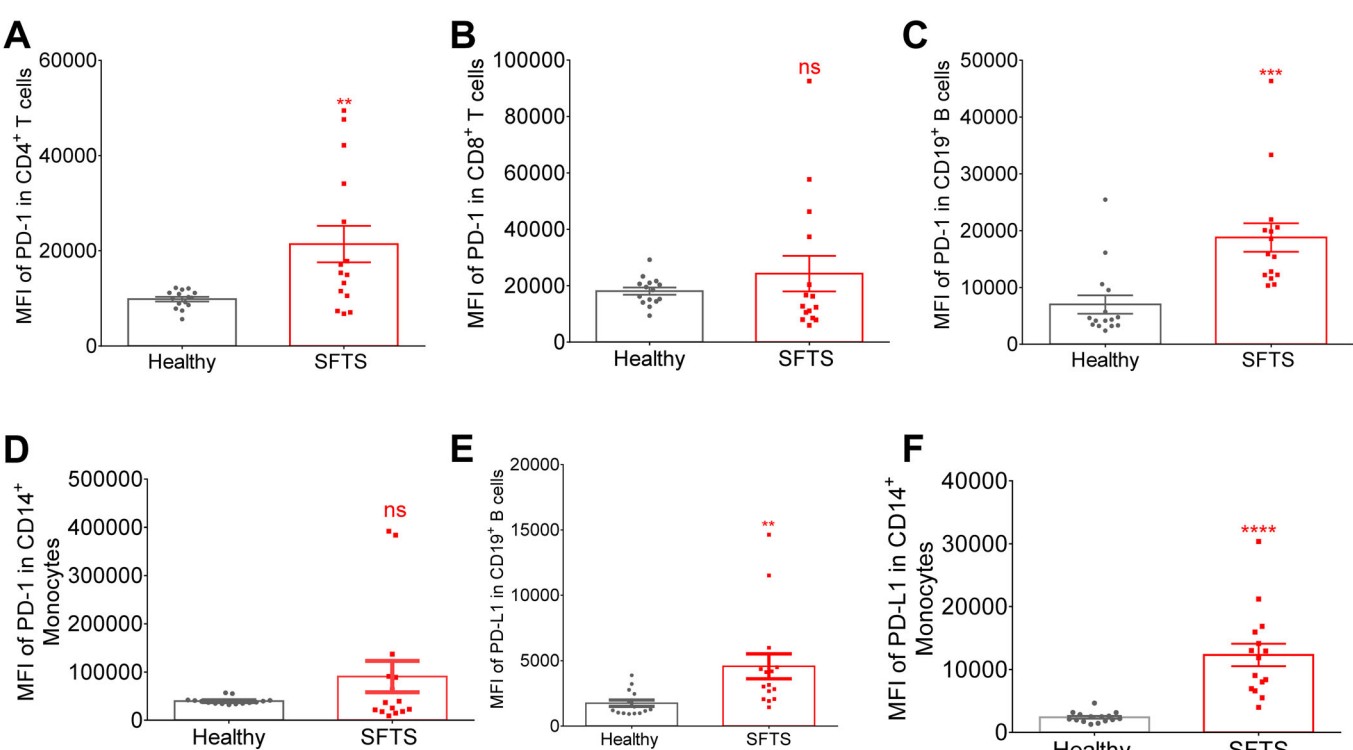

**Figure EV1.** **The expression levels of PD-1/PD-L1 was upregulated in immune cells of SFTS patients.**

(**A–D**) The expression levels of PD-1 in CD4[+] (**A**) and CD8[+] (**B**) T, CD19[+] B (**C**) cells and CD14[+] monocytes (**D**) was summarized for the SFTS patients (*n* = 15) and the healthy control (*n* = 15). Two-tailed unpaired *t* test was performed to compare SFTS patients with healthy control (ns, no significance; **P = 0.0058; ***P = 0.0005). (**E, F**) The expression levels of PD-L1 in CD19[+] B cells (**E**) and CD14[+] monocytes (**F**) was summarized for the SFTS patients (*n* = 15) and the healthy control (*n* = 15). Two-tailed unpaired *t* test was performed to compare SFTS patients with healthy control (**P = 0.0079; ****P < 0.0001). Data information: (**A–F**) data are shown as mean ± SEM. ns, no significance; **P < 0.01; ***P < 0.001; ****P < 0.0001.

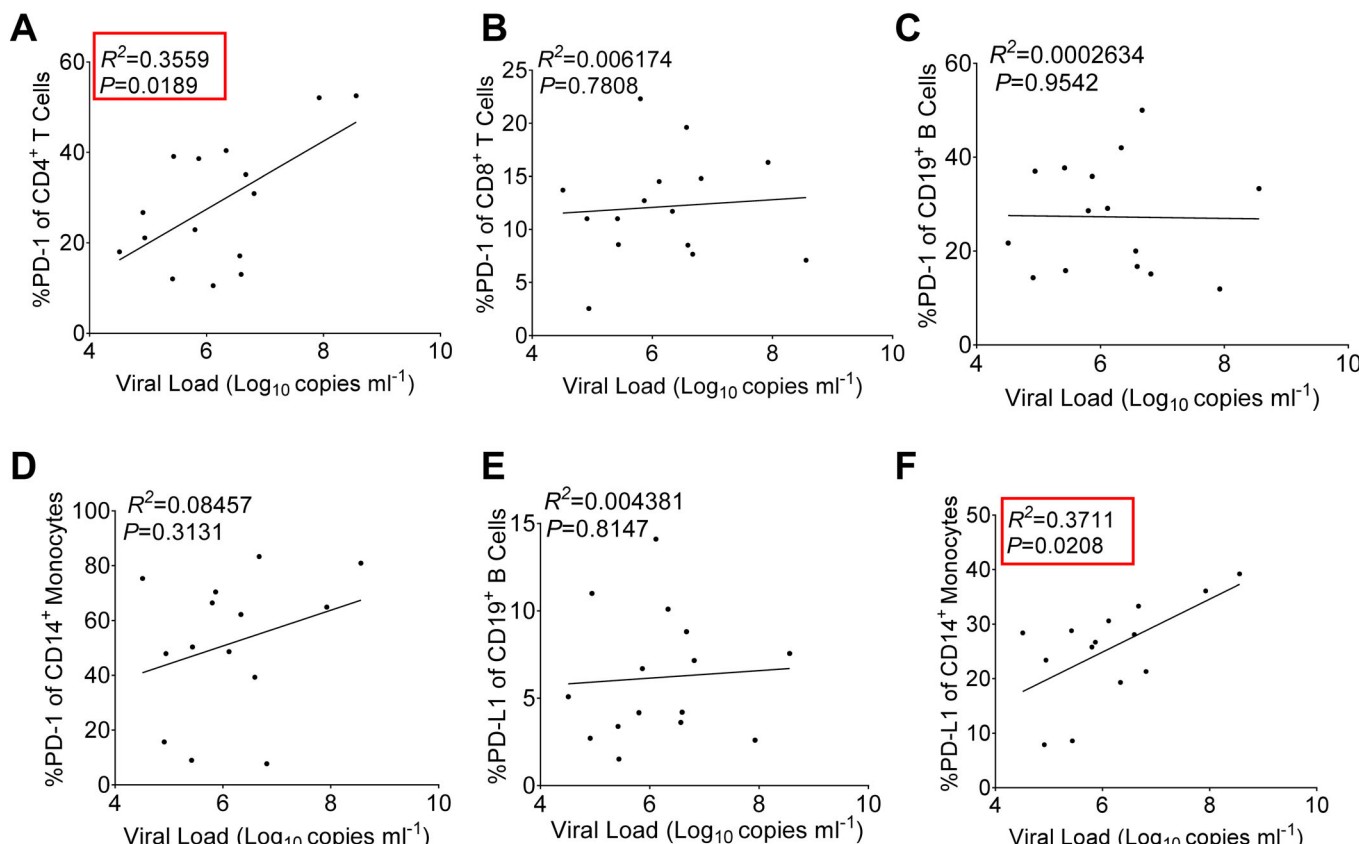

**Figure EV2.   Correlation between PD-1/PD-L1 expression and serum viral load in SFTS patients.**

(A–D) Correlation between PD-1 in CD4$^+$ (A) and CD8$^+$ (B) T, CD19$^+$ B (C) cells and CD14$^+$ monocytes (D) expression and serum viral load in SFTS patients. (E, F) Correlation between PD-L1 in CD19$^+$ B cells (E) and CD14$^+$ monocytes (F) and viral RNA copies in serum viral load in SFTS patients. Data information: Correlation analyses were performed by linear regression using the GraphPad Prism 6.0 program, Pearson's correlation tests were used to measure the strength of association between variables.

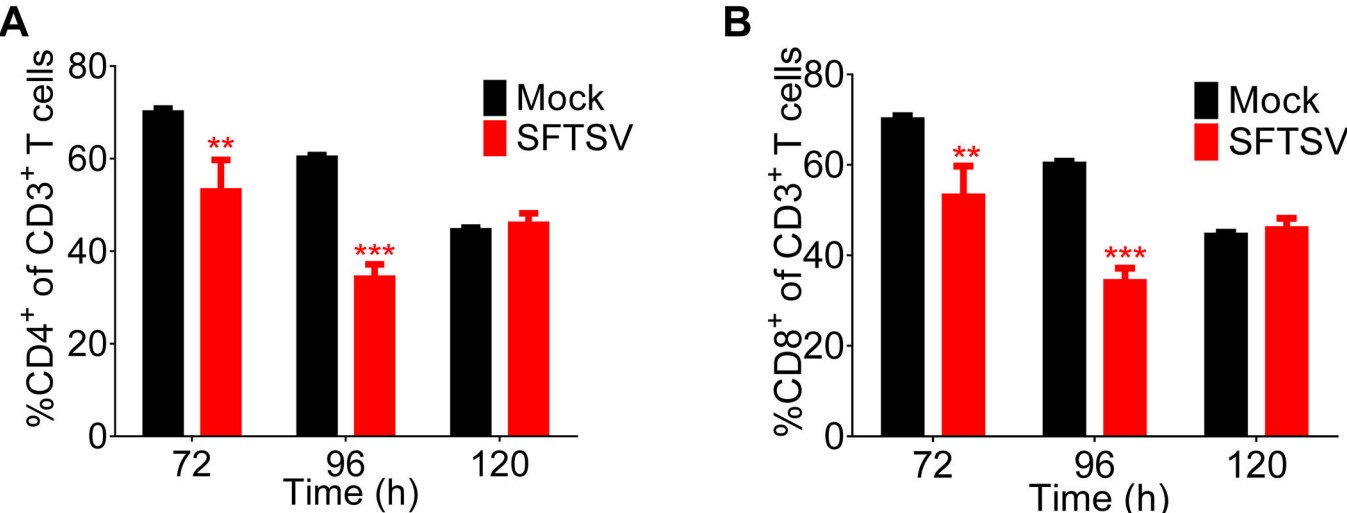

**Figure EV3. Kinetics of the T/B lymphocytes during SFTSV-infected PBMC.**

(A, B) The expression of CD4$^+$ (A) and CD8$^+$ (B) T cells was summarized for the uninfected controls ($n = 3$) and the SFTSV (MOI $= 1$) infection ($n = 3$) at 72/96/120 h. Two-way ANOVA with Sidak's multiple comparisons test was performed to compare SFTSV infection with uninfected control (**$P = 0.0053$; ***$P = 0.0001$). Data information: Data are shown as mean ± SEM. *ns*, no significance; **$P < 0.01$; ***$P < 0.001$. The *n* means the numbers of sample repeats in one experiment at a same time.

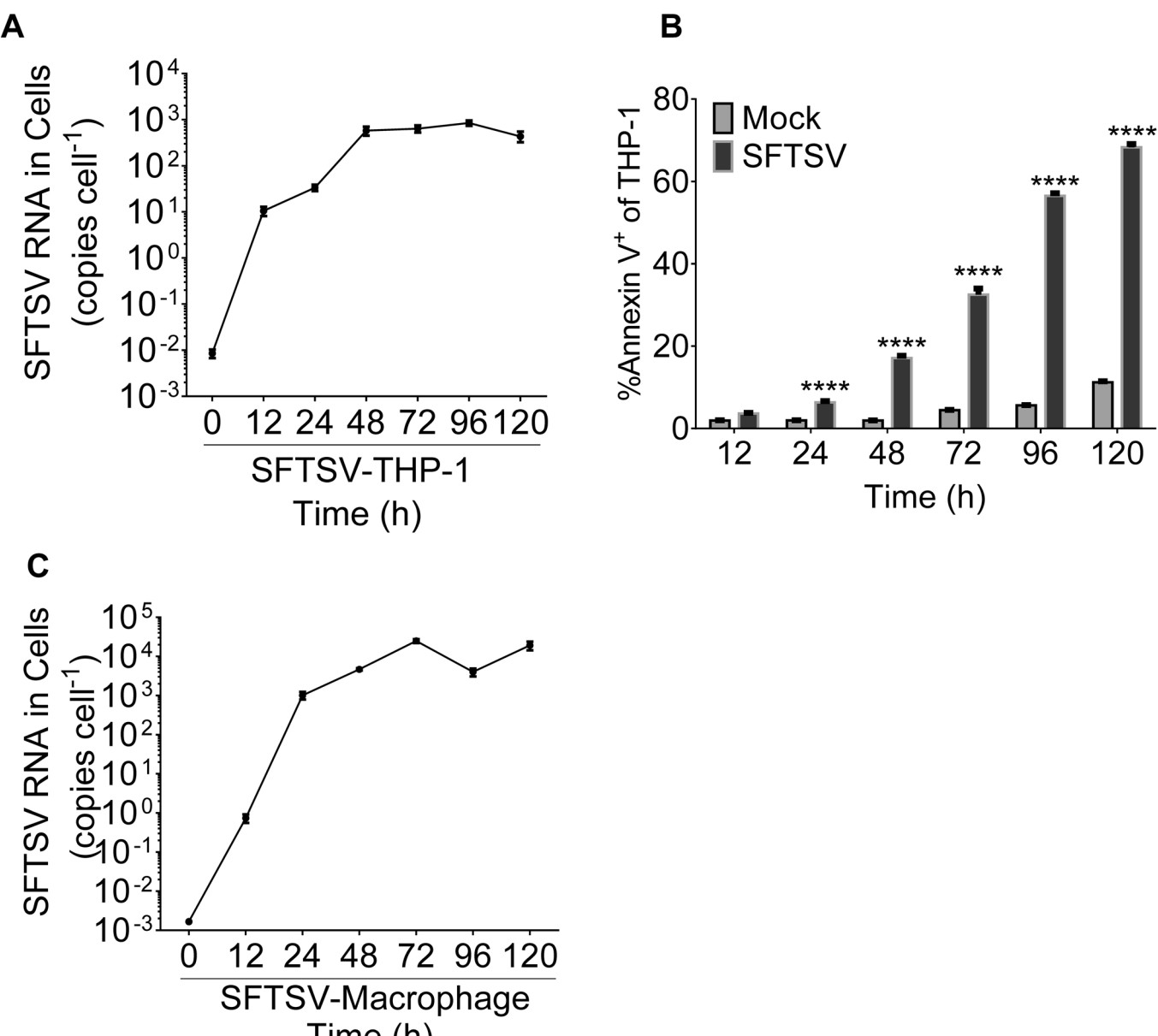

**Figure EV4. Kinetics of viral replication in THP-1/Macrophage cells was examined by serial sampling of cells.**

(A) Kinetics of viral replication in THP-1 cells was examined at SFTSV (MOI = 1) infection ($n = 3$). (B) Annexin V$^+$ expression of THP-1 cells was summarized for the uninfected controls ($n = 3$) and the SFTSV (MOI = 1) infection ($n = 3$). Two-way ANOVA with Sidak's multiple comparisons test was performed to compare SFTSV infection with uninfected control (****$P < 0.0001$). (C) Kinetics of viral replication in macrophage cells was examined at SFTSV (MOI = 1) infection ($n = 3$). Data information: Data are shown as mean ± SEM. ****$P < 0.0001$. The $n$ means the numbers of sample repeats in one experiment at a same time.

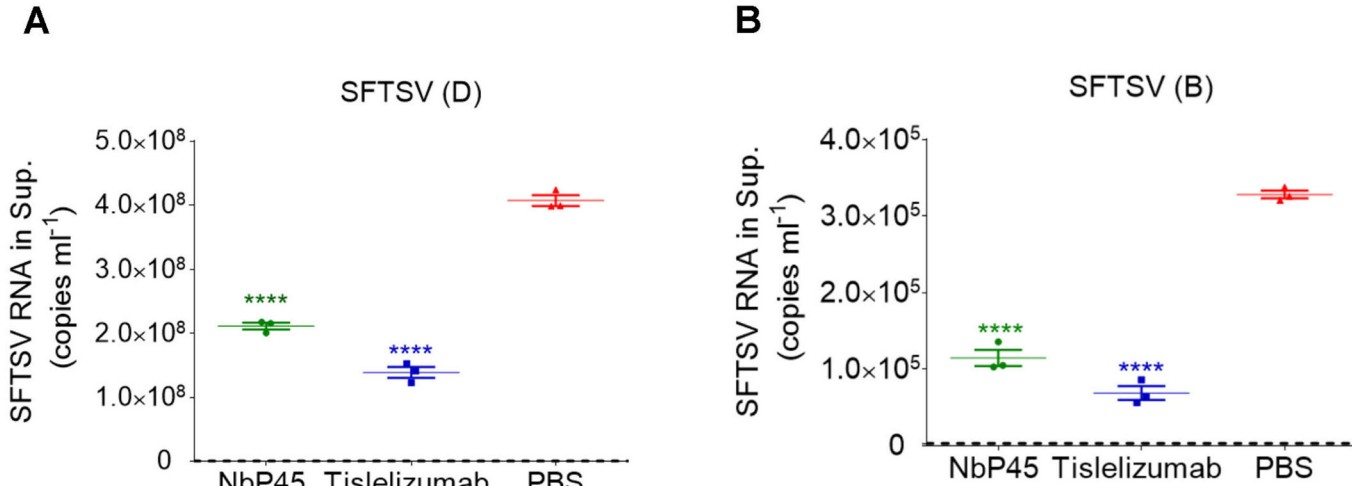

**Figure EV5.  NbP45 potently inhibited another strain of SFTSV replication.**

(A, B). The inhibition activity of NbP45 ($n = 3$) or Tislelizumab ($n = 3$) against SFTSV of subtype D (A) or subtype B (B) (MOI = 1) infection PBMCs at 48 hpi. The black dashed line was non-infected control. One-way ANOVA with Tukey's test was performed to compare treatment group with control group (PBS) (****$P < 0.0001$). Data information: Data are shown as mean ± SEM. ****$P$ <0.0001. The $n$ means the numbers of sample repeats in one experiment at a same time.

