## [Peer Review File · EMBO Molecular Medicine]

Inhibition of SFTSV replication in humanized mice by a subcutaneously administered anti-PD1 nanobody

Mengmeng Ji, Jiaqian Hu, Doudou Zhang, Bilian Huang, Shijie Xu, Na Jiang, Yuxin Chen, Yujiong Wang, Xilin Wu and Zhiwei Wu

Corresponding authors: Zhiwei Wu (wzhw@nju.edu.cn), Xilin Wu (xilinwu@nju.edu.cn), Yujiong Wang (wyj@nxu.edu.cn) and Yuxin Chen (yuxin.chen@nju.edu.cn)

Review Timeline:

Submission Date:	4th Sep 23
Editorial Decision:	29th Sep 23
Revision Received:	12th Dec 23
Editorial Decision:	9th Jan 24
Revision Received:	15th Jan 24
Accepted:	17th Jan 24

Editor: Zeljko Durdevic

Transaction Report:

29th Sep 2023

Dear Dr. Wu,

Thank you for the submission of your manuscript to EMBO Molecular Medicine. We have now received feedback from the three reviewers who agreed to evaluate your manuscript. All three referees recognize potential interest of the study but also raise serious concerns that should be addressed in a major revision. If you would like to discuss further the points raised by the referees, I am available to do so via email or video. Let me know if you are interested in this option.

We would welcome the submission of a revised version within three months for further consideration. Please let us know if you require longer to complete the revision.

I look forward to receiving your revised manuscript.

Yours sincerely,

Zeljko Durdevic

We require:

- 1) A .docx formatted version of the manuscript text (including legends for main figures, EV figures and tables). Please make sure that the changes are highlighted to be clearly visible.
- 2) Individual production quality figure files as .eps, .tif, .jpg (one file per figure). For guidance, download the 'Figure Guide PDF': (<https://www.embopress.org/page/journal/17574684/authorguide#figureformat>).
- 3) A .docx formatted letter INCLUDING the reviewers' reports and your detailed point-by-point responses to their comments. As part of the EMBO Press transparent editorial process, the point-by-point response is part of the Review Process File (RPF), which will be published alongside your paper.
- 4) A complete author checklist, which you can download from our author guidelines (<https://www.embopress.org/page/journal/17574684/authorguide#submissionofrevisions>). Please insert information in the checklist that is also reflected in the manuscript. The completed author checklist will also be part of the RPF.
- 5) Please note that all corresponding authors are required to supply an ORCID ID for their name upon submission of a revised manuscript.
- 6) It is mandatory to include a 'Data Availability' section after the Materials and Methods. Before submitting your revision, primary datasets produced in this study need to be deposited in an appropriate public database, and the accession numbers and

database listed under 'Data Availability'. Please remember to provide a reviewer password if the datasets are not yet public (see <https://www.embopress.org/page/journal/17574684/authorguide#dataavailability>).

13) Author contributions: You will be asked to provide CRediT (Contributor Role Taxonomy) terms in the submission system. These replace a narrative author contribution section in the manuscript.

14) A Conflict of Interest statement should be provided in the main text.

15) Every published paper now includes a 'Synopsis' to further enhance discoverability. Synopses are displayed on the journal

webpage and are freely accessible to all readers. They include a short stand first (maximum of 300 characters, including space) as well as 2-5 one-sentences bullet points that summarizes the paper. Please write the bullet points to summarize the key NEW findings. They should be designed to be complementary to the abstract - i.e. not repeat the same text. We encourage inclusion of key acronyms and quantitative information (maximum of 30 words / bullet point). Please use the passive voice. Please attach these in a separate file or send them by email, we will incorporate them accordingly.

Please note: When submitting your revision you will be prompted to enter your funding and payment information. This will allow Wiley to send you a quote for the article processing charge (APC) in case of acceptance. This quote takes into account any reduction or fee waivers that you may be eligible for. Authors do not need to pay any fees before their manuscript is accepted and transferred to the publisher.

EMBO Press participates in many Publish and Read agreements that allow authors to publish Open Access with reduced/no publication charges. Check your eligibility: <https://authorservices.wiley.com/author-resources/Journal-Authors/open-access/affiliation-policies-payments/index.html>

***** Reviewer's comments *****

Referee #1 (Remarks for Author):

The manuscript submitted by Mengmeng Ji et al entitled " Inhibition of SFTSV replication in humanized mice by a subcutaneously administered anti-PD1 nanobody " aims to uncover that the PD-1 blockade is a potential therapeutic strategy against SFTSV replication. An anti-PD1 nanobody, NbP45, effectively inhibited SFTSV infection in PBMCs by reducing apoptosis and enhancing T lymphocyte proliferation. The manuscript is also very well-written and structured. However, there are several aspects that need to be improved to make the conclusions more convincing. As the follows:

Major points:

1. Figure 2. It is suggested that the effect of SFTSV infection on lymphocyte apoptosis and the effect of viral infection on lymphocyte apoptosis after down-regulation of PD-1 should be added to the Figures or supplementary materials. Although PD-1 is a gene involved in apoptosis. However, the regulation of SFTSV infection on apoptosis is a multi-factor synthesis process. The apoptosis phenotypes in this research system still need to be shown to be more convincing.
2. Figure 4. Which domain of PD-1 does NbP45 bind to? Is it through this domain that PD-1 and pD-L1 are prevented from binding?
3. Figure 5A, C and D. The level of T lymphocyte apoptosis was inhibited by NbP45 by nearly 2.5 times, but the proliferation level of the virus was only decreased by 1.5 times, and the inhibitory effect shown here is a statistical asterisk, which is not very significant. The results of in vivo experiments showed that the virus dropped nearly 20 times after NbP45 treatment (Fig. 7B). Please explain these differences.
4. The logic of this study is that NbP45 inhibits the apoptosis of T lymphocytes by binding to PD-1, promotes the proliferation of T cells, and then inhibits the replication of SFTSV. But the mechanism of this process is unclear. The researchers showed the changes of PD-1 expression level during viral infection, but they did not show the effect of PD-1 on T lymphocyte apoptosis in this study system. In fact, the mechanism by which NbP45 inhibits apoptosis needs to be further explored. The immune escape mechanism of the virus remains to be studied. Whether NbP45 inhibits viral replication through other pathways is not clear. Therefore, the tone of this study needs to be softened (e.g. Lines 264-266).
5. Line 735. The uniform writing is "PD-L1".

Referee #2 (Remarks for Author):

In this manuscript, authors demonstrated the PD-1/PD-L1 pathway involved in the acute SFTSV infection, and found the potent inhibitory effect of novel anti-PD-1 nanobody, NbP45, against SFTSV infection in vitro and in SFTSV-infected NCG-HuPBL mouse model. However, the following comments and questions need to be addressed before publication.

1. As mentioned in author's previous study (Xu et al., PLoS Pathog, 17(5): e1009587, 2021), severe clinical symptoms or death appears in NCG-HuPBL mice after SFTSV infection. It would be helpful if authors provide mouse weight and survival curve results that are crucial for evaluating the antiviral efficacy in this study.
2. In vitro and in vivo experiments have confirmed that both NbP45 and licensed anti-PD-1 antibody Tislelizumab can inhibit

virus replication and improve immune cell function. Please try to explain why Tislelizumab cannot alleviate SFTSV induced thrombocytopenia like NbP45 in vivo.

3. To evaluate the in vivo efficacy of NbP45, it is necessary to supplement the count and function (apoptosis, proliferation and activation) of various immunocytes (T, B cells and monocytes), especially need to supplement the data of antibody responses against SFTSV antigens (Gn and/or NP) in plasma.

4. The author explored the PD-1/PD-L1 pathway serve as potential therapeutic targets for SFTSV infection. The potential therapeutic effects of other immune checkpoint molecules, such as CTLA-4 and TIGIT, in SFTSV infection need to be discussed.

5. The author's study indicates that SFTSV infection can induce upregulation of PD-1/PD-L1, leading to abnormal function of immunocytes and impairing virus clearance, indirectly promoting virus replication. However, in this study, author need to pay attention to whether PD-1/PD-L1 has a direct impact on virus entry into target cells and virus replication. Because immune checkpoint molecule NRP-1, was shown to serve as an entry factor and potentiate SARS-CoV-2 infectivity (Daly JL et al., Science, 370(6518): 861-865, 2020).

6. Fig. 5: It will be more appropriate to use NB15 instead of PBS treatment for the control group. Fig. 7B, 7C: NbP45 (s.c.) should be compared with NB15 (s.c.) rather than PBS group, and the data comparison between NbP45 (s.c.) and control (NB15) group in Fig. 7B should be clearly described in figure legend.

7. Several clades (or clusters) exist in SFTSV and some studies showed that the morbidity and mortality in epidemic areas are different according to the genotype of SFTSV. The author should provide a brief introduction to the pathogenicity and geographical distribution characteristics of SFTSV genotypes E (JS-2013-14), D, and B involved in this study. And provide virus detailed information (Genebank number) on Materials and Methods section.

8. Severe fever with thrombocytopenia syndrome virus (SFTSV) officially named Dabie bandavirus. Authors should put this in the upper part of the introduction.

9. There are many minor mistakes in your manuscript:

Results, Discussion, Materials and Methods section: Please note there is a space between numbers and units.

Line 237: TCID₅₀ should be written in standard, "50" requires subscripts.

Referee #3 (Comments on Novelty/Model System for Author):

Reviewer's comments

General comments

The present study reports the involvement of the PD-1/PD-L1 pathway during acute SFTSV infection and suggest its potential as a host target for immunotherapy interventions against SFTSV infection. Moreover, the authors identified a new anti-PD1 nanobodies (NbP45), which was isolated from an alpaca immunized with human PD-1. They also confirmed that NbP45 can effectively inhibited SFTSV infection in PBMCs by reducing apoptosis and enhancing T lymphocyte proliferation, and showed better efficacy compared to a licensed anti-PD1 antibody in an SFTSV-infected humanized mouse model. Overall, the study's design exhibits both reasonability and innovation, thereby holding significant promise for the clinical implementation of SFTSV.

Referee #3 (Remarks for Author):

Defects:

1. Page 3, line 232. I am interested in ascertaining the distribution and titer of the virus within the tissue of both the NbP45 administration group and the control group in the mice model. These data are expected to facilitate a more comprehensive analysis of the drug's inhibitory effect.

2. The writer should check the grammar and other errors of the sentences in the MS. A more concise and logical expression would be better for research publication.

Point-to-point answers to reviewers' comments

***** Reviewer's comments *****

Referee #1 (Remarks for Author):

The manuscript submitted by Mengmeng Ji et al entitled " Inhibition of SFTSV replication in humanized mice by a subcutaneously administered anti-PD1 nanobody " aims to uncover that the PD-1 blockade is a potential therapeutic strategy against SFTSV replication. An anti-PD1 nanobody, NbP45, effectively inhibited SFTSV infection in PBMCs by reducing apoptosis and enhancing T lymphocyte proliferation. The manuscript is also very well-written and structured. However, there are several aspects that need to be improved to make the conclusions more convincing. As the follows:

Reply: We sincerely appreciate the positive feedback from the reviewer on our manuscript. The reviewer pointed out several issues and gave a number of suggestions, which are very helpful for us to improve the manuscript. According to these valuable advices, we have amended the relevant parts of the manuscript and addressed the questions raised by the reviewer.

Major points:

1. Figure 2. It is suggested that the effect of SFTSV infection on lymphocyte apoptosis and the effect of viral infection on lymphocyte apoptosis after down-regulation of PD-1 should be added to the Figures or supplementary materials. Although PD-1 is a gene involved in apoptosis. However, the regulation of SFTSV infection on apoptosis is a multi-factor synthesis process. The apoptosis phenotypes in this research system still need to be shown to be more convincing.

Reply: The reviewer's concern is well taken. To include the effect of SFTSV infection on lymphocyte apoptosis and the impact of viral infection on lymphocyte apoptosis after down-regulating PD-1 in Figure 2 or supplementary materials, we conducted additional experiments to address this concern.

We evaluated the influence of SFTSV infection on PBMCs apoptosis and the impact of viral infection on lymphocyte apoptosis after down-regulating PD-1. PBMCs derived from healthy individuals were infected with SFTSV (subtype E, JS-2013-14) at an MOI of 1 for 120 hours. PD-1 down-regulation was achieved by using NbP45 or Tislelizumab blockade (Fig 4F).

The results revealed a significant increase in the expression of annexin V⁺ in

CD4⁺/CD8⁺ T cells in the infected samples (PBS) at 72/96/120 hours post-infection compared to the uninfected control (mock). Notably, blocking PD-1 with NbP45 or Tislelizumab led to a significant downregulation of annexin V⁺ expression in CD4⁺/CD8⁺ T cells compared to the infected control (PBS). Concurrently, the data demonstrated a substantial inhibition of SFTSV replication by blocking PD-1 with NbP45 or Tislelizumab, resulting in a reduction of viral RNA copies by more than 1.0×10^7 copies/ml (Appendix Figure S1).

In summary, these additional experiments support and illustrate the impact of SFTSV infection on lymphocyte apoptosis and the subsequent modulation of lymphocyte apoptosis following down-regulation of PD-1.

Appendix Figure S1. Apoptosis of the T lymphocytes during SFTSV infected PBMC.

A, B. Annexin V⁺ expression of CD4⁺ (A) and CD8⁺ (B) T cell was summarized for the uninfected ($n=3$), SFTSV (MOI=1) infected ($n=3$) and treated controls ($n=3$) at 72/96/120 h.

C. The inhibitory activity of NbP45 ($n=3$) or Tislelizumab ($n=3$) against SFTSV (MOI=1) infection of PBMCs at 72/96/120 h. Each line represents data from a group with indicated treatment.

Data are shown as mean \pm SD. Two-way ANOVA with Tukey's test was performed to compare treatment group with control group. *ns*, no significance; * $p < 0.05$; ** $p < 0.01$; *** $p < 0.001$; **** $p < 0.0001$.

2. Figure 4. Which domain of PD-1 does NbP45 bind to? Is it through this domain that PD-1 and PD-L1 are prevented from binding?

Reply: Our western-blot analysis demonstrated a NbP45 binding preference for PD-1 under non-reduced conditions compared to reduced conditions (Fig 4B). This suggests that NbP45 recognizes a conformational epitope. Moreover, corroborating evidence of this binding was observed through ELISA, BLI, and FACS assays (Fig 4C, D, E). To assess whether NbP45 could impede the interaction between PD-1 and PD-L1, we conducted an inhibition assay using FACS. The results indicated a dose-dependent inhibition by NbP45 against PD-1 binding with PD-L1 (Fig 4F). These findings collectively suggest that NbP45 binding domain overlap with that of the PD-L1 binding PD-1. It is necessary to conduct point mutation analysis of PD-1 or crystal structural analysis of PD-1-NbP45 complex to illustrate specific mechanisms of NbP45 inhibition.

3. Figure 5A, C and D. The level of T lymphocyte apoptosis was inhibited by NbP45 by nearly 2.5 times, but the proliferation level of the virus was only decreased by 1.5 times, and the inhibitory effect shown here is a statistical asterisk, which is not very significant. The results of in vivo experiments showed that the virus dropped nearly 20 times after NbP45 treatment (Fig. 7B). Please explain these differences.

Reply: We appreciate the reviewer's insightful comments on Figure 5A, C, and D, highlighting the discrepancy between the reduced levels of T lymphocyte apoptosis and viral proliferation after NbP45 treatment.

Regarding the first question: Our findings suggest a positive correlation between T lymphocyte apoptosis and decreased viral titer in the supernatant. While the reduction of virus in the supernatant is slightly lower compared to T lymphocyte levels, we propose that SFTSV primarily infects macrophages, monocytes, or B cells rather than T cells (Suzuki *et al*, 2020; Xu *et al*, 2021; Zhang *et al*, 2019). Consequently, NbP45 inhibits virus replication by binding to PD-1 on T cells, activating them to eliminate SFTSV-infected cells and resulting in a reduction of viral load in the supernatant. This selective roles of PD-1 may account for the discrepancy observed.

Regarding the second question: In our in vivo experiments evaluating NbP45 efficacy against SFTSV infection in humanized NCG-HuPBL mice (Fig 7A), where human T cells are detectable, we observed a gradual reduction in viral load over the course of infection (Fig 7B). Notably, the significant difference in viral loads between the treatment and control (PBS) groups becomes apparent at 9, 12, and 15 days post-infection. The mice exhibited a reduction in viremia of over 2 log₁₀ at 9 days post-infection, followed by a reduction of 1 log₁₀ at both 12 and 15 days post-infection. These results indicate that viral load inhibition requires time to

manifest, unlike the in vitro setting where the proliferation level of the virus was only decreased by 1.5 times 48 hours after NbP45 treatment. This discrepancy between in vitro and in vivo observations is consistent with similar phenomena reported in anti-PD1 antibodies inhibiting HIV-1 infection (Bobardt *et al*, 2020; Calvet-Mirabent *et al*, 2022; Harper *et al*, 2020).

4. The logic of this study is that NbP45 inhibits the apoptosis of T lymphocytes by binding to PD-1, promotes the proliferation of T cells, and then inhibits the replication of SFTSV. But the mechanism of this process is unclear. The researchers showed the changes of PD-1 expression level during viral infection, but they did not show the effect of PD-1 on T lymphocyte apoptosis in this study system. In fact, the mechanism by which NbP45 inhibits apoptosis needs to be further explored. The immune escape mechanism of the virus remains to be studied. Whether NbP45 inhibits viral replication through other pathways is not clear. Therefore, the tone of this study needs to be softened (e.g. Lines 264-266).

Reply: We appreciate the insightful comments by the reviewer regarding the logic and mechanisms presented in our study. Taking these valuable suggestions into consideration, we have made necessary revisions to improve the clarity and tone of our manuscript. For instance, on Page 2, Lines 32-33, we added the statement, "potentially achieved through the mitigation of apoptosis and the augmentation of T lymphocyte proliferation." Additionally, on Page 14, Lines 263, we incorporated the phrase, "PD-1 could down-regulate T cell activity during immune responses in order to prevent autoimmune tissue damage."

5. Line 735. The uniform writing is "PD-L1".

Reply: Thank you for pointing out the inconsistency. We have thoroughly reviewed the manuscript and corrected the errors to ensure uniform usage of "PD-L1" throughout the revised version.

Referee #2 (Remarks for Author):

In this manuscript, authors demonstrated the PD-1/PD-L1 pathway involved in the acute SFTSV infection, and found the potent inhibitory effect of novel anti-PD-1 nanobody, NbP45, against SFTSV infection in vitro and in SFTSV-infected NCG-HuPBL mouse model. However, the following comments and questions need to be addressed before publication.

Reply: We sincerely appreciate the reviewer for the valuable comments and questions, which have significantly enhanced the quality of our manuscript.

1. As mentioned in author's previous study (Xu *et al.*, PLoS Pathog, 17(5): e1009587, 2021), severe clinical symptoms or death appears in NCG-HuPBL mice after SFTSV

infection. It would be helpful if authors provide mouse weight and survival curve results that are crucial for evaluating the antiviral efficacy in this study.

Reply: In our previous study (Xu *et al.*, 2021), we noted that SFTSV-infected animals experienced 10–30% weight loss by day 28 compared to the non-infected group. Higher virus titers correlated with increased mortality rates. In our current study, we recorded mouse weight and monitored survival (Appendix Figure S2). Notably, NbP45 treatment ameliorated this weight loss (Appendix Figure S2A).

It's essential to acknowledge the unique challenges posed by the NCG-HuPBL mouse model, where Graft-versus-host disease (GvHD) induced by the interaction of human T cells with mouse tissues can contribute to weight loss and mortality (Kosugi-Kanaya *et al.*, 2017). Although we initially omitted body weight and survival rate data due to GvHD's influence, we now include these parameters in our manuscript to provide a comprehensive evaluation of the antiviral efficacy.

Appendix Figure S2. Evaluation of NbP45 therapeutic efficacy in SFTSV-infected NCG-HuPBL mice.

A, B. Relative weight (A) and survival (B) were assessed and are shown as standard error of the mean. Each line represents data from 1 group.

C-F. WBCs (C), neutrophils (D), lymphocytes (E), and monocytes (F)

were analyzed in 7 groups of NCG-HuPBL mice infected with SFTSV at 15 days. The normal range is between the 2 dashed lines. Each dot represents data from 1 mouse.

Data are shown as mean \pm SD. Two-way ANOVA with Tukey's test was performed to compare treatment group with control group (PBS). *ns*, no significance; * $p < 0.05$; ** $p < 0.01$; *** $p < 0.001$; **** $p < 0.0001$.

2. In vitro and in vivo experiments have confirmed that both NbP45 and licensed anti-PD-1 antibody Tislelizumab can inhibit virus replication and improve immune cell function. Please try to explain why Tislelizumab cannot alleviate SFTSV induced thrombocytopenia like NbP45 in vivo.

Reply: Previous studies have highlighted the variable efficacy of different anti-PD1 antibodies with respect to factors such as epitope specificity, affinity, pharmacokinetics (PK), and pharmacodynamics (PD) (Cowles *et al*, 2022). Both Tislelizumab, a whole IgG, and NbP45, a nanobody, demonstrated similar potency in vitro but surprisingly exhibited divergent effects in vivo.

To explore the underlying reasons, we conducted pharmacokinetic analyses in a mouse model. Subcutaneous administration revealed that NbP45 achieved approximately double the area under the curve (AUC) compared to Tislelizumab (Fig 6A, B). Furthermore, tissue distribution studies exhibited robust fluorescence signals in the lung, liver, kidneys, and stomach after 336 hours with subcutaneous administration of NbP45 (Fig 6C, D). Additionally, serum antibody concentration gradually decreased with subcutaneous NbP45 administration compared to Tislelizumab (Fig 7E).

These findings suggest that NbP45 exhibits superior pharmacokinetics via subcutaneous administration, potentially explaining why it outperformed Tislelizumab in alleviating SFTSV-induced thrombocytopenia in vivo.

3. To evaluate the in vivo efficacy of NbP45, it is necessary to supplement the count and function (apoptosis, proliferation and activation) of various immunocytes (T, B cells and monocytes), especially need to supplement the data of antibody responses against SFTSV antigens (Gn and/or NP) in plasma.

Reply: In accordance with prior studies (Wu *et al*, 2020; Xu *et al.*, 2021), we evaluated NbP45 in vivo efficacy against SFTSV infection in NCG-HuPBL mice by assessing viral load, platelet counts, and leukocyte populations, including WBCs, neutrophils, lymphocytes, and monocytes (Appendix Figure S2C-F). Notably, NbP45-treated mice showed no significant leukocytosis compared to controls (Appendix Figure S2C-F).

In our established SFTSV-infected NCG-HuPBL mouse model, human cells were activated by the foreign mouse environment, leading to proliferation and activation of various immunocytes (T, B cells, and monocytes). Therefore, we initially did not study their function (apoptosis, proliferation, and activation) as our prior data

indicated their activation by the foreign environment.

To address reviewer's concerns about antibody responses against SFTSV antigens (Gn and/or NP) in plasma, we only measured antibodies against SFTSV antigens, considering severely immunodeficient nature of NCG mice, as characterized by T-cell, B-cell, and NK cell defects. As anticipated, there was no detectable antibody response to Gn or NP protein (data not shown).

Appendix Figure S2. Evaluation of Nbp45 therapeutic efficacy in SFTSV-infected NCG-HuPBL mice.

A, B. Relative weight (A) and survival (B) were assessed and are shown as standard error of the mean. Each line represents data from 1 group.

C-F. WBCs (C), neutrophils (D), lymphocytes (E), and monocytes (F) were analyzed in 7 groups of NCG-HuPBL mice infected with SFTSV at 15 days. The normal range is between the 2 dashed lines. Each dot represents data from 1 mouse.

Data are shown as mean \pm SD. Two-way ANOVA with Tukey's test was performed to compare treatment group with control group (PBS). *ns*, no significance; * $p < 0.05$; ** $p < 0.01$; *** $p < 0.001$; **** $p < 0.0001$.

4. The author explored the PD-1/PD-L1 pathway serve as potential therapeutic targets

for SFTSV infection. The potential therapeutic effects of other immune checkpoint molecules, such as CTLA-4 and TIGIT, in SFTSV infection need to be discussed.

Reply: The reviewer's suggestion is well taken and relative discussion has been incorporated into our manuscript. We have discussed the potential therapeutic effects of other immune checkpoint molecules, including CTLA-4, CD47, PD-L1, and TIGIT, in the discussion section of the revised manuscript (lines 342-359).

5. The author's study indicates that SFTSV infection can induce upregulation of PD-1/PD-L1, leading to abnormal function of immunocytes and impairing virus clearance, indirectly promoting virus replication. However, in this study, author need to pay attention to whether PD-1/PD-L1 has a direct impact on virus entry into target cells and virus replication. Because immune checkpoint molecule NRP-1, was shown to serve as an entry factor and potentiate SARS-CoV-2 infectivity (Daly JL et al., *Science*, 370(6518): 861-865, 2020).

Reply: The reviewer's comment is well taken and we performed additional experiment to address the reviewer's concern. To investigate the potential direct impact of PD-1 on SFTSV entry and replication, we evaluated whether NbP45 could neutralize SFTSV infection by binding to PD-1. Our results revealed that NbP45 could bind to PD-1 on THP-1 cells (Appendix Figure S3A), while it did not exhibit any inhibitory effect on SFTSV infection in the absence of PBMCs (Appendix Figure S3B). In contrast, neutralizing control antibodies (Anti-sGn Nb) displayed a dose-dependent inhibition against SFTSV infection (Appendix Figure S3B). These findings suggest that NbP45 does not directly inhibit SFTSV infection by binding to PD-1 on infected cells.

Appendix Figure S3. Characterization of NbP45.

A. Histogram depicting the NbP45 or Tislelizumab binding to PD-1 of THP-1 cells, as evaluated using flow cytometry. Nb Ctl (one published nanobody, Nb15, specific for SARS-CoV-2 S protein) served as negative nanobody control.

B. Neutralization activity of NbP45 against live SFTSV infection of PBMCs. Anti-sGn Nb (one nanobody, specific for SFTSV Gn protein), Nb Ctl (one published nanobody, Nb15, specific for SARS-CoV-2 S protein) served as positive and negative nanobody control, respectively.

Data are shown as mean \pm SD.

6. Fig. 5: It will be more appropriate to use NB15 instead of PBS treatment for the control group. Fig. 7B, 7C: NbP45 (s.c.) should be compared with NB15 (s.c.) rather than PBS group, and the data comparison between NbP45 (s.c.) and control (NB15) group in Fig. 7B should be clearly described in figure legend.

Reply: The reviewer's suggestion is well taken and we have revised accordingly.

7. Several clades (or clusters) exist in SFTSV and some studies showed that the morbidity and mortality in epidemic areas are different according to the genotype of SFTSV. The author should provide a brief introduction to the pathogenicity and geographical distribution characteristics of SFTSV genotypes E (JS-2013-14), D, and B involved in this study. And provide virus detailed information (Genbank number) on Materials and Methods section.

Reply: We appreciate the reviewer's insightful suggestion, and have incorporated a brief introduction to the pathogenicity, geographical distribution and characteristics of SFTSV genotypes E (JS-2013-14), D, and B based on published references, particularly Hu et al., Li et al. and Yun et al. (Hu *et al*, 2023; Li *et al*, 2017; Yun *et al*, 2020). Additionally, in response to the reviewer's recommendation, we have included the specific virus information in the Materials and Methods section (lines 485-486 and 493-494) for clarity:

JS-2013-24 (subtype E) (GenBank accession no. AMY99382)

20ZJTZ07 (subtype D) (GenBank accession no. UNP61901)

HB29 (Subtype B) (GenBank accession no. NC_018137)

8. Severe fever with thrombocytopenia syndrome virus (SFTSV) officially named Dabie bandavirus. Authors should put this in the upper part of the introduction.

Reply: We appreciate the reviewer's suggestion and have made such a description as suggested. In the Introduction on page 3, we have included a sentence stating that Severe Fever with Thrombocytopenia Syndrome Virus (SFTSV) is officially named Dabie bandavirus (lines 42-44).

9. There are many minor mistakes in your manuscript:

Results, Discussion, Materials and Methods section: Please note there is a space between numbers and units.

Reply: We thank the reviewer for pointing out the errors and have done the corrections accordingly.

Line 237: TCID₅₀ should be written in standard, "50" requires subscripts.

Reply: We appreciate the reviewer's attention to details, and we have corrected the formatting of TCID₅₀.

Referee #3 (Comments on Novelty/Model System for Author):

Reviewer's comments

General comments

The present study reports the involvement of the PD-1/PD-L1 pathway during acute SFTSV infection and suggest its potential as a host target for immunotherapy interventions against SFTSV infection. Moreover, the authors identified a new anti-PD1 nanobodies (NbP45), which was isolated from an alpaca immunized with human PD-1. They also confirmed that NbP45 can effectively inhibited SFTSV infection in PBMCs by reducing apoptosis and enhancing T lymphocyte proliferation, and showed better efficacy compared to a licensed anti-PD1 antibody in an SFTSV-infected humanized mouse model. Overall, the study's design exhibits both reasonability and innovation, thereby holding significant promise for the clinical implementation of SFTSV.

Reply: We express our gratitude to reviewer for the insightful comments and constructive feedback. The thoughtful assessment has significantly contributed to the enhancement of our manuscript.

Referee #3 (Remarks for Author):

Defects:

1. Page 3, line 232. I am interested in ascertaining the distribution and titer of the virus within the tissue of both the NbP45 administration group and the control group in the mice model. These data are expected to facilitate a more comprehensive analysis of the drug's inhibitory effect.

Reply: In our previous study (Xu *et al.*, 2021), we established an SFTSV-infected NCG-HuPBL mice model that replicates key pathological features observed in human SFTSV infection. Our investigations demonstrated the presence of viral RNA in various tissues, including blood, heart, liver, spleen, lungs, kidneys, stomach, intestine, brain, and muscles. Notably, immunofluorescence (IFC) results indicated the spleen as a prominent site of virus detection, indicating consistency between viral RNA copies in blood and various tissues. Additionally, a reduction in platelets emerged as a primary clinical symptom.

Building upon these insights, our current study focuses on assessing viral RNA, conducting blood routine tests, and monitoring body weight. These parameters, validated as reliable indices of efficacy in our earlier work on antibody treatment for SFTSV infection in NCG-HuPBL mice (Wu *et al.*, 2020), form the foundation for a comprehensive analysis of the inhibitory effect of NbP45 administration in the mouse model.

2. The writer should check the grammar and other errors of the sentences in the MS. A more concise and logical expression would be better for research publication.

Reply: We appreciate the reviewer's feedback, and we have thoroughly checked the manuscript, implementing numerous changes to enhance language and clarity. The revised sections are highlighted in RED throughout the manuscript.

References:

- Bobardt M, Kuo J, Chatterji U, Wiedemann N, Vuagniaux G, Gallay P (2020) The inhibitor of apoptosis proteins antagonist Debio 1143 promotes the PD-1 blockade-mediated HIV load reduction in blood and tissues of humanized mice. *PLoS One* 15: e0227715
- Calvet-Mirabent M, Sánchez-Cerrillo I, Martín-Cófreces N, Martínez-Fleta P, de la Fuente H, Tsukalov I, Delgado-Arévalo C, Calzada MJ, de Los Santos I, Sanz J *et al* (2022) Antiretroviral therapy duration and immunometabolic state determine efficacy of ex vivo dendritic cell-based treatment restoring functional HIV-specific CD8+ T cells in people living with HIV. *EBioMedicine* 81: 104090
- Cowles SC, Sheen A, Santollani L, Lutz EA, Lax BM, Palmeri JR, Freeman GJ, Wittrup KD (2022) An affinity threshold for maximum efficacy in anti-PD-1 immunotherapy. *MAbs* 14: 2088454
- Harper J, Gordon S, Chan CN, Wang H, Lindemuth E, Galardi C, Falcinelli SD, Raines SLM, Read JL, Nguyen K *et al* (2020) CTLA-4 and PD-1 dual blockade induces SIV reactivation without control of rebound after antiretroviral therapy interruption. *Nat Med* 26: 519-528
- Hu J, Li W, Peng Z, Chen Z, Shi Y, Zheng Y, Liang Q, Wu Y, Liu W, Shen W *et al* (2023) Annual incidence and fatality rates of notifiable infectious diseases in southeast China from 1950 to 2022 and relationship to socioeconomic development. *J Glob Health* 13: 04107
- Kosugi-Kanaya M, Ueha S, Abe J, Shichino S, Shand FHW, Morikawa T, Kurachi M, Shono Y, Sudo N, Yamashita A *et al* (2017) Long-Lasting Graft-Derived Donor T Cells Contribute to the Pathogenesis of Chronic Graft-versus-Host Disease in Mice. *Front Immunol* 8: 1842
- Li Z, Hu J, Cui L, Hong Y, Liu J, Li P, Guo X, Liu W, Wang X, Qi X *et al* (2017) Increased Prevalence of Severe Fever with Thrombocytopenia Syndrome in Eastern China Clustered with Multiple Genotypes and Reasserted Virus during 2010-2015. *Sci Rep* 7: 6503

Suzuki T, Sato Y, Sano K, Arashiro T, Katano H, Nakajima N, Shimojima M, Kataoka M, Takahashi K, Wada Y *et al* (2020) Severe fever with thrombocytopenia syndrome virus targets B cells in lethal human infections. *J Clin Invest* 130: 799-812

Wu X, Li Y, Huang B, Ma X, Zhu L, Zheng N, Xu S, Nawaz W, Xu C, Wu Z (2020) A single-domain antibody inhibits SFTSV and mitigates virus-induced pathogenesis in vivo. *JCI Insight* 5

Xu S, Jiang N, Nawaz W, Liu B, Zhang F, Liu Y, Wu X, Wu Z (2021) Infection of humanized mice with a novel phlebovirus presented pathogenic features of severe fever with thrombocytopenia syndrome. *PLoS Pathog* 17: e1009587

Yun S-M, Park S-J, Kim Y-I, Park S-W, Yu M-A, Kwon H-I, Kim E-H, Yu K-M, Jeong HW, Ryou J *et al* (2020) Genetic and pathogenic diversity of severe fever with thrombocytopenia syndrome virus (SFTSV) in South Korea. *JCI Insight* 5

Zhang L, Fu Y, Wang H, Guan Y, Zhu W, Guo M, Zheng N, Wu Z (2019) Severe Fever With Thrombocytopenia Syndrome Virus-Induced Macrophage Differentiation Is Regulated by miR-146. *Front Immunol* 10: 1095

9th Jan 2024

Dear Dr. Wu,

Thank you for the submission of your revised manuscript to EMBO Molecular Medicine. I am pleased to inform you that we will be able to accept your manuscript pending the following final amendments:

1) Figures: Main figures and EV figures should be uploaded as individual high-resolution files, with their legends in the manuscript text. Please check "Author Guidelines" for more information:

<https://www.embopress.org/page/journal/17574684/authorguide#figureformat>

<https://www.embopress.org/page/journal/17574684/authorguide#expandedview>

2) Tables: Table EV1 should be uploaded as a separate file and its legend should be removed from the manuscript and added to the table.

3) We note that you currently have together with you, a total of 3 co-corresponding authors. Is that correct? Do you confirm equal contribution of these authors, able to take full responsibility for the paper and its content? While there is no limit per se to the number of co-corresponding authors, 3 co-corresponding authors is rather rare, and may not reflect as intended to the community.

4) In the main manuscript file, please do the following:

- Please address all comments suggested by our data editors listed below:

o Figure legends:

1. Please note that the 'Data Information' section is not labelled in the legends of figures 1a, c-h; 2; 3; 4; 5; 6; 7; 8; EV 1; EV 2; EV 3; EV 4; EV 5.

2. Please indicate the statistical test used for data analysis in the legends of figures EV 2a-f.

3. Please note that in figures 1a, c-h; 3a, c, e, g; 7b-d; EV 1a-f; EV 3a-b; EV 4b; EV 5a-b, there is a mismatch between the annotated p values in the figure legend and the annotated p values in the figure file that should be corrected.

4. Please note that the box plots need to be defined in terms of minima, maxima, centre, bounds of box and whiskers, and percentile in the legends of figures 1a; 3a; EV 1a-f.

5. Please note that information related to n is missing in the legends of figures 1a; 4d-e; 5b; 6f; 7b-d.

6. Although 'n' is provided, please describe the nature of entity for 'n' in the legends of figures 2a-e; 3a, c, e; g; 5a, c-j; EV 3a-b; EV 4a-c; EV 5a-b.

7. Please note that the error bars are not defined in the legends of figures 4d-e.

- In M&M, provide the antibody dilutions that were used for each antibody.

- In M&M, provide the statement that in addition to the WMA Declaration of Helsinki the experiments also conformed to the principles set out in the Department of Health and Human Services Belmont Report.

- In M&M, a statistical paragraph should reflect all information that you have filled in the Authors Checklist, especially regarding randomization, blinding, replication.

- Author contributions: Please remove it from the manuscript and specify author contributions in our submission system. CRediT has replaced the traditional author contributions section because it offers a systematic machine-readable author contributions format that allows for more effective research assessment. You are encouraged to use the free text boxes beneath each contributing author's name to add specific details on the author's contribution. More information is available in our guide to authors:

<https://www.embopress.org/page/journal/17574684/authorguide#authorshipguidelines>

5) Appendix: Please add a table of content with page numbers and move the appendix figure legends under each corresponding figure.

6) Funding: Please make sure that information about all sources of funding are complete in both our submission system and in the manuscript. The Major Research and Development Project (2018ZX10301406), and Nanjing University-Ningxia University Collaborative Project (Grant# 2017BN04) are missing in our submission system.

7) The Paper Explained: Please add it to the main manuscript file.

8) Synopsis:

- Synopsis image: Please provide the visual abstract as a high-resolution .jpeg file 550 px-wide x (250-400)-px high.

9) For more information: This space should be used to list relevant web links for further consultation by our readers. Could you identify some relevant ones and provide such information as well? Some examples are patient associations, relevant databases, OMIM/proteins/genes links, author's websites, etc...

10) As part of the EMBO Publications transparent editorial process initiative (see our Editorial at

<http://embomolmed.embopress.org/content/2/9/329>), EMBO Molecular Medicine will publish online a Review Process File (RPF) to accompany accepted manuscripts. This file will be published in conjunction with your paper and will include the anonymous referee reports, your point-by-point response and all pertinent correspondence relating to the manuscript. Let us know whether you agree with the publication of the RPF and as here, if you want to remove or not any figures from it prior to publication. Please note that the Authors checklist will be published at the end of the RPF.

11) Please provide a point-by-point letter INCLUDING my comments as well as the reviewer's reports and your detailed

responses (as Word file).

I look forward to reading a new revised version of your manuscript as soon as possible.

Yours sincerely,

Zeljko Durdevic

*** Instructions to submit your revised manuscript ***

1) a .docx formatted version of the manuscript text (including Figure legends and tables)

2) Separate figure files*

3) supplemental information as Expanded View and/or Appendix. Please carefully check the authors guidelines for formatting Expanded view and Appendix figures and tables at <https://www.embopress.org/page/journal/17574684/authorguide#expandedview>

4) a letter INCLUDING the reviewer's reports and your detailed responses to their comments (as Word file).

5) The paper explained: EMBO Molecular Medicine articles are accompanied by a summary of the articles to emphasize the major findings in the paper and their medical implications for the non-specialist reader. Please provide a draft summary of your article highlighting

6) For more information: There is space at the end of each article to list relevant web links for further consultation by our readers. Could you identify some relevant ones and provide such information as well? Some examples are patient associations, relevant databases, OMIM/proteins/genes links, author's websites, etc...

7) Author contributions: the contribution of every author must be detailed in a separate section.

8) EMBO Molecular Medicine now requires a complete author checklist (<https://www.embopress.org/page/journal/17574684/authorguide>) to be submitted with all revised manuscripts. Please use the checklist as guideline for the sort of information we need WITHIN the manuscript. The checklist should only be filled with page numbers where the information can be found. This is particularly important for animal reporting, antibody dilutions (missing) and

exact values and n that should be indicated instead of a range.

9) Every published paper now includes a 'Synopsis' to further enhance discoverability. Synopses are displayed on the journal webpage and are freely accessible to all readers. They include a short stand first (maximum of 300 characters, including space) as well as 2-5 one sentence bullet points that summarise the paper. Please write the bullet points to summarise the key NEW findings. They should be designed to be complementary to the abstract - i.e. not repeat the same text. We encourage inclusion of key acronyms and quantitative information (maximum of 30 words / bullet point). Please use the passive voice. Please attach these in a separate file or send them by email, we will incorporate them accordingly.

You are also welcome to suggest a striking image or visual abstract to illustrate your article. If you do please provide a jpeg file 550 px-wide x 300-800px high.

10) A Conflict of Interest statement should be provided in the main text

11) Please note that we now mandate that all corresponding authors list an ORCID digital identifier. This takes <90 seconds to complete. We encourage all authors to supply an ORCID identifier, which will be linked to their name for unambiguous name identification.

Currently, our records indicate that the ORCID for your account is 0000-0002-0672-948X.

Link Not Available

Photos 400-800 DPI

*Additional important information regarding figures and illustrations can be found at <https://bit.ly/EMBOPressFigurePreparationGuideline>. See also figure legend preparation guidelines: <https://www.embopress.org/page/journal/17574684/authorguide#figureformat>

***** Reviewer's comments *****

Referee #1 (Comments on Novelty/Model System for Author):

The authors have addressed all the questions.

Referee #1 (Remarks for Author):

The paper is suggested to be published in EMBO Molecular Medicine.

Referee #2 (Remarks for Author):

The author provided a clear and detailed explanation of the review comments. No further questions, agree to accept.

The authors addressed the minor editorial issues.

17th Jan 2024

Dear Dr. Wu,

We are pleased to inform you that your manuscript is accepted for publication and is now being sent to our publisher to be included in the next available issue of EMBO Molecular Medicine.
